# Stimulus uncertainty and relative reward rates determine adaptive responding in perceptual decision-making

Luis de la Cuesta-Ferrer[1], Christina Koß [2], Sarah Starosta[3], Nils Kasties[3], Daniel Lengersdorf[3], Frank Jäkel[2], Maik C. Stüttgen[1]*

**1** Institute of Pathophysiology, University Medical Center of the Johannes Gutenberg University Mainz, Mainz, Germany, **2** Centre for Cognitive Science, Institute of Psychology, Technical University of Darmstadt, Darmstadt, Germany, **3** Department of Biopsychology, Faculty of Psychology, Ruhr University Bochum, Bochum, Germany

* maik.stuettgen@uni-mainz.de

## Abstract

In dynamic environments, animals must select actions based on sensory input as well as expected positive and negative consequences. This type of behavior is typically studied using perceptual decision making (PDM) tasks. The arguably most influential framework for describing the cognitive processes underlying PDM is signal detection theory (SDT). One central assumption of SDT is that observers make perceptual decisions by comparing sensory evidence to a static decision criterion. However, mounting evidence suggests that the criterion is in fact highly dynamic and that observers adjust it flexibly according to task demands. Nevertheless, the mechanisms by which observers integrate stimulus and reward information for adaptive criterion learning remain not well understood. Here, we systematically investigated the factors influencing criterion setting at the single-trial level. To that end, we first specified three SDT-based models that learn either from reward, reward omission, or both. Next, by concomitantly manipulating stimulus and reward probabilities, we constructed experimental conditions in which these models make divergent predictions. Finally, we subjected rats and pigeons to a PDM task comprising these conditions. We find that subjects adopted decision criteria that maximize total reward in all experimental conditions. Detailed behavioral analyses reveal that criterion learning is driven by the integration of rewards, not reward omissions, and that reward integration is influenced by two additional factors: first, the degree of stimulus uncertainty, and second, the difference in the relative reward rates (rather than the absolute reward rates) between the choice alternatives. A model incorporating these factors accounts well for criterion dynamics across experimental conditions for both species and links signal detection theory to a learning mechanism operating at the level of single trials which, in the steady state, produces behavior similar to the matching law, a central tenet of learning theory.

License, which permits unrestricted use, distribution, and reproduction in any medium, provided the original author and source are credited.

**Data availability statement:** All relevant data and analysis codes are available under https://github.com/ldelacue/de_la_Cuesta_2024.

**Funding:** Preparation of this work was supported by grants from the Deutsche Forschungsgemeinschaft to M.C.S. (project IDs 197059818, 424828846, and 543128862) and F.J. (project ID 424828846). The funders had no role in study design, data collection and analysis, decision to publish, or preparation of the manuscript.

**Competing interests:** The authors have declared that no competing interests exist.

## Author summary

Humans and other animals rely on their senses and experience to categorize objects and pursue their goals. For example, a mushroom hunter uses sight, smell and touch as well as knowledge of the local biota to decide whether to pick a particular mushroom. The consequences of erring can be dire – food intoxication if savoring a poisonous exemplar, or a meager dinner if too many palatable mushrooms are rejected. Also, the hunter's decision may be influenced by ambient lighting conditions or his estimate of how likely it is to encounter poisonous mushrooms in a particular area at a particular time of year. Our work is concerned with the algorithms that animals use to make such decisions and how they adapt when circumstances, such as stimulus discriminability, change. We show that mathematical models that incorporate the animals' uncertainty about the type of stimulus currently being perceived make very similar decisions to the animals. Furthermore, we find that the animals balance their choices by considering relative, rather than absolute, reward expectations, reflecting a long-standing principle of animal learning theory. Together, these features collectively allow animals to obtain nearly as many rewards as theoretically possible.

## Introduction

Perceptual decision-making (PDM) is the process of using sensory information to select an appropriate course of action [1]. For example, an animal overhearing the cracking of branches in the woods will act differently depending on whether it interprets the sound as indicating the approach of a conspecific or a predator. In controlled laboratory settings, PDM is studied using experimental tasks wherein subjects are required to map (usually two) well-defined responses to two or more well-controlled stimuli. Such experiments have been used extensively to probe the limits of sensory detection and discrimination capabilities in humans and animals [2–5] and to identify lawful relationships between physical stimulus properties, subjective sensations, and their neural substrates [6–12]. Arguably the most influential framework for understanding and modeling processes of PDM is signal detection theory (SDT) [13]. We will here provide a brief introduction to the SDT framework, which we will later extend to investigate how subjects make adaptive perceptual decisions; see [14] for a detailed outline.

Consider the case of a two-stimulus, two-response discrimination task. On each trial, one of two stimuli is presented to the observer, with equal probability, and the observer is asked to indicate which stimulus is present. SDT provides a model of the covert decision process of the observer. First, SDT in its most common form assumes that repeated presentations of the same stimulus generate random values on a decision axis according to a normal distribution (Fig 1A). In a single trial, the subject is thus confronted with a random value x drawn from one of the two equal-variance normal distributions, each corresponding to one of two stimuli (S1 or S2). The subject has to decide which distribution this value was sampled from. The distance between

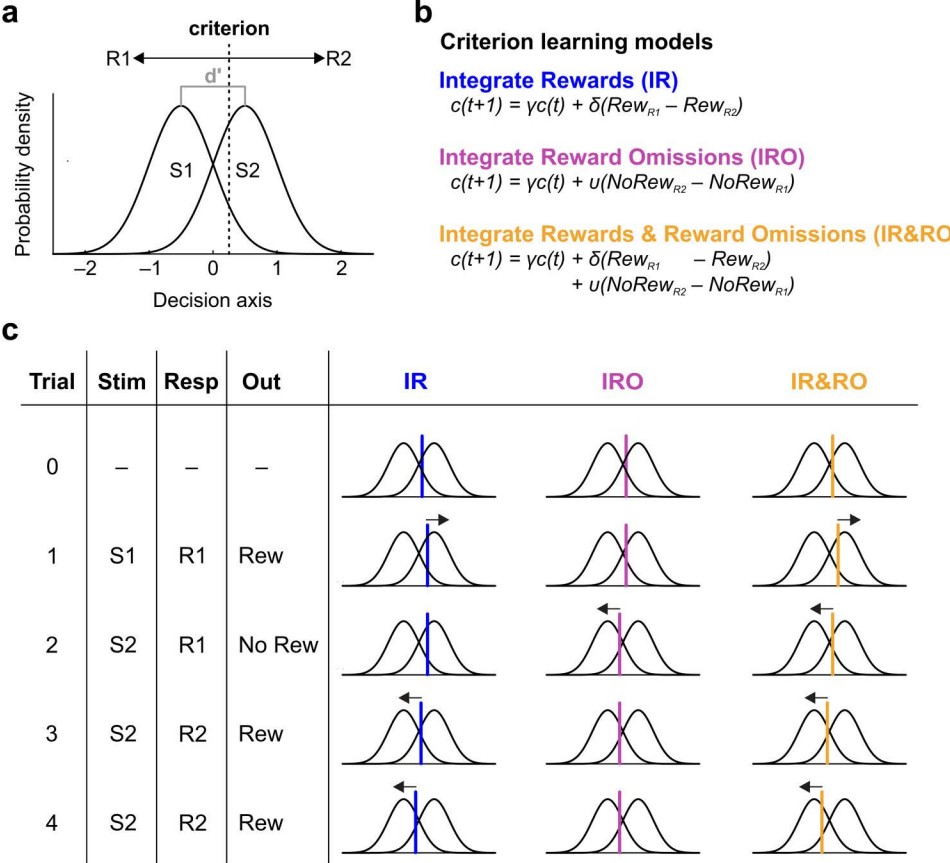

**Fig 1. Adaptive decision-making in a detection-theory framework.** **a.** Illustration of SDT for a two-stimulus, two-response conditional discrimination task. On any given trial, presentation of a stimulus is tantamount to drawing a random sample x from either of the two distributions (S1 and S2), giving rise to a specific value of a decision variable. The distance between the means of the two distributions (expressed in units of standard deviations) is called "sensitivity" or d' (gray). The subject makes the decision to emit either response (R1 or R2) by comparing x to a criterion c (vertical dotted line): R1 if c < x, R2 if c > x. In classical SDT, the value of the criterion is fixed, but the value of x changes from trial to trial. In this example, d' = 2 and c = 0.25. If S1 and S2 are presented equally often, expected accuracy (fraction correct trials) is 0.83 across both stimuli (0.89 in S1 trials and 0.77 in S2 trials). **b.** Definitions of three mechanistic models of adaptive criterion setting: 1) Integrate Rewards (IR, blue), 2) Integrate Reward Omissions (IRO, purple), 3) Integrate Rewards and Reward Omissions (IR&RO, yellow). Basically, all models specify that the criterion in the next trial t+1 equals the criterion in the present trial t times the leak factor γ, and c is incremented or decremented by δ depending on whether the response was rewarded or not. **c.** Exemplification of the criterion updating mechanisms of the three models in a sequence of four consecutive trials. Following stimulus presentation, the subject compares the current value of the decision variable generated by stimulus presentation x with the current value of criterion c, emits the selected response, receives a reward or not, and then shifts the criterion in accordance with the update rule of the specific model. The IR model shifts the criterion after each reward, the IRO model after each reward omission, and the IR&RO model after both rewards and reward omissions.

the means of the two distributions, divided over their standard deviation is called 'sensitivity' or d' and serves as a measure of sensory discriminability. The subject indicates the decision by emitting one of two corresponding responses (R1 for S1 and R2 for S2). SDT assumes this decision is taken by comparing the likelihood ratio of the two stimuli against a decision criterion. In the equal-variance normal case, this boils down to comparing the observed value of the decision variable x with a fixed decision criterion c – if x < c, the subject emits R1, when x ≥ c, the subject emits R2. (Importantly, the criterion is not to be confused with a perceptual threshold; in SDT, no such threshold exists.) Thus, SDT separates the perceptual decision of the subject into two independent processes: the perception of the stimulus per se, and the decision of the subject to assign it to one of two mutually exclusive categories. Moreover, the theory provides two distinct metrics

for these processes – d' for sensitivity, criterion for a non-sensory response bias, both of which can be easily calculated from choice data [15].

SDT has overturned classic beliefs about the existence of sensory thresholds [16]. Many of its basic assumptions have been supported in a great number of studies [14,17–19], and the theory is continually being tested and extended [20–27]. Also, concepts from SDT are widely used in neuroscience, e.g., to compare neurometric and psychometric stimulus discrimination performance [8,9] and to separate the effects of stimulus and bias on neuronal activity [28–32].

One crucial assumption of SDT is that the observer maintains a static criterion within each experimental condition. Making this assumption is necessary for the calculation of d' and c from aggregated choice data. However, there are many reasons to question its validity. First, there exists notable trial-to-trial variability in criterion (e.g., [24,33,34]); second, observers' responses are usually autocorrelated, even for perfectly randomized stimulus presentations (e.g., [35–37]). Nonetheless, SDT neither specifies any criterion updating mechanism nor does it explicitly acknowledge criterion variability at all, although unaccounted criterion variability biases the estimates of d' and criterion (and therefore threshold and slopes of psychometric functions, thereby affecting the validity of SDT's most central assets [38–40]. Importantly, criterion variability is of high theoretical relevance because it offers a way to understand trial history effects for preceding stimuli, choices, and outcomes, and provides a window to understand the cognitive processes underlying both perceptual processing as well as more generally behavioral adaptation to novel contingencies [41–49]. Moreover, multiple experimental manipulations reliably affect the decision criterion (e.g., [50–52]), but SDT does not propose any rule that specifies which criterion will be chosen in a given experimental condition, let alone any mechanism how the criterion changes when an experimental variable is manipulated.

The inadequacy of a static criterion and the importance of understanding perceptual decisions at a trial-by-trial level was acknowledged early on [53] and remains so to this day [38]. Nonetheless, there have been only few attempts to accommodate criterion variability and the effects of experimental manipulations on criterion setting [14]. Specifically, we know of no systematic attempts to compare models featuring different criterion update rules 1) in their ability to fit trial-by-trial choice data, 2) with respect to their steady-state criterion predictions, as well as 3) their ability to generate qualitatively similar choice responses in forward simulations [23,51,54–64]. Beyond addressing an important shortcoming of SDT, a thoroughly tested trial-by-trial criterion learning model would allow to correlate fluctuating criterion levels to measures of neural activity, as has been successfully accomplished for action values derived from reinforcement learning models in dynamic foraging tasks (e.g., [65,66]).

The motivation of this study was to study and develop different models of criterion setting with the objective of describing adaptive responding in various experimental conditions. Building on previous work [55,57,67–69], we first specified three SDT-based trial-by-trial criterion learning models. Next, by concomitantly manipulating discrimination difficulty, stimulus presentation probabilities, and reward probabilities, we constructed experimental conditions in which these models make diverging predictions. We then subjected rats to a PDM task comprising these conditions to assess the models' performance against empirical data with respect to their predictions for steady-state criteria, trial-by-trial fits, and ability to reproduce the subjects' behavior in forward simulations.

We found that subjects managed to maximize rewards in all experimental conditions. Model comparisons revealed that obtained rewards, rather than reward omissions, drives criterion learning. Moreover, our findings indicate that the degree of criterion updating is determined by the current trial's stimulus' uncertainty. Additionally, we demonstrate that the steady-state criterion is independent of the total amount of reward received from both sides (i.e., global reward rate or reward density) and only depends on the relative reward rates. This connects to the matching law, a well-established finding from animal learning theory [70,71] which however is rarely accounted for in the field of perceptual decision-making (but see [62,63,72]). Finally, we demonstrate that a reward-integration model incorporating these features is capable of fitting and reproducing the behavioral data observed in a second, similarly structured experiment which was conducted with pigeons as subjects.

## Results

### Modeling adaptive perceptual decisions in a detection-theory framework

SDT itself is silent as to how specific decision criteria are selected or learned. However, one can extend this framework by specifying how the criterion changes trial by trial as a function of the preceding sequence of stimuli, responses and outcomes. Here, we initially consider three criterion-setting models embodying three straightforward learning rules (Fig 1B). Reflecting their mechanistic structures, these models are named 1) Integrate Rewards (IR), 2) Integrate Reward Omissions (IRO), and 3) Integrate Rewards & Reward Omissions (IR&RO). The operation of these models is exemplified for a series of 5 trials in Fig 1C. A learning agent operating under the IR model only shifts the criterion on rewarded trials (in Fig 1C, trials 1, 3 and 4) as to make the rewarded response more likely to occur in the subsequent trial. In contrast, an agent operating under the IRO model shifts the criterion only in unrewarded trials as to reduce the likelihood of emitting the unsuccessful response again (in Fig 1C, only in trial 2). And third, an agent operating under the IR&RO model shifts the response criterion in both rewarded and unrewarded trials. The models feature one or two learning rates which control the size of the update steps: $\delta$ for updating after reward in IR and IR&RO models, and $\upsilon$ (upsilon) for updating after reward omissions in IRO and IR&RO. Additionally, all models feature a leaky integration of past criteria whose extent is controlled by the leak term $\gamma$ (the effect of $\gamma$ is to pull $c$ towards 0 in each trial, its effect is not shown in Fig 1C for simplicity; see Methods for a more detailed explanation).

We started with these three models because their trial-by-trial learning rules are arguably among the most basic conceivable in a detection theory framework and have, in a similar form, been proposed earlier [55,67] (we added the leak term $\gamma$ which is needed to prevent exclusive choice when integrating rewards).

### Design of the experimental conditions and behavioral task structure

The three models not only specify exact learning rules for trial-by-trial updating of the decision criterion, but also allow to derive equilibrium predictions for the steady state given a set of parameters (stimulus means, $\gamma$, $\delta$, and $\upsilon$; see Methods for details). We accordingly designed a set of experimental conditions in which the three models predict qualitatively different steady-state criterion locations. Additionally, we calculated the reward-maximizing (henceforth, optimal) criterion location for each condition as a benchmark.

The general principle of how we designed our experimental conditions is outlined in Fig 2A. Generally speaking, we started with a set of five distinct stimuli. These stimuli are customized for each experimental subject such that their perceptual distances are similar, and that most stimuli are to a certain extent confusable. For each condition, three or four stimuli are chosen, and each stimulus is assigned to one of two mutually exclusive categories. Then, we specify each stimulus' presentation probability as well as the reward probability (note that only correct responses are rewarded). Through the combination of these manipulations – stimulus selection, category assignment, specification of presentation and reward probabilities – we constructed several experimental scenarios (henceforth, conditions) in which the three models introduced above make divergent predictions as to where the criterion should be located in the steady state (i.e., after subjects had a reasonable amount of experience with that condition).

Fig 2B presents a general overview of the experimental conditions, Table 1 details stimulus sets, categories, and presentation and reward probabilities for each condition, S1 Fig provides a more detailed illustration of how the predictions were generated for each model, and S2 Fig explains how the stimulus set for each subject was chosen. Generally speaking, there were three pairs of conditions dubbed "Rich", "Lean", and "Confuse". Each of these conditions came in two varieties, L and R. This nomenclature (left, L, and right, R) follows from the criterion shift that would be expected in each of these conditions from an optimal account. "Left" implies an increase in the frequency of R2, "right" implies an increase in the frequency of R1, as illustrated in Fig 1A. L and R varieties of each condition were constructed by mirroring the stimulus locations and their associated reward and presentation probabilities at the baseline category boundary (Fig 2B). Crucially, conditions Rich and Lean are scaled versions of each other in terms of the relative amounts of reward to be expected from the two categories (i.e., differing only in reward density or global reward rate), whereas the Confuse condition features

### a  Construction of experimental conditions (example: Lean L)

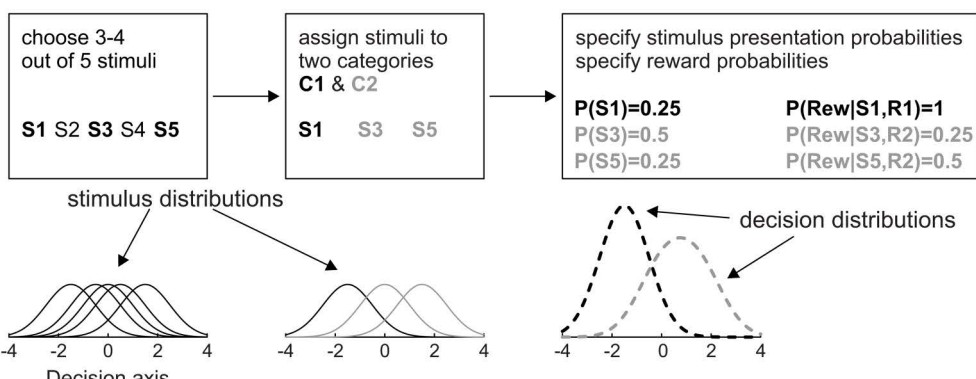

### b  Experimental conditions

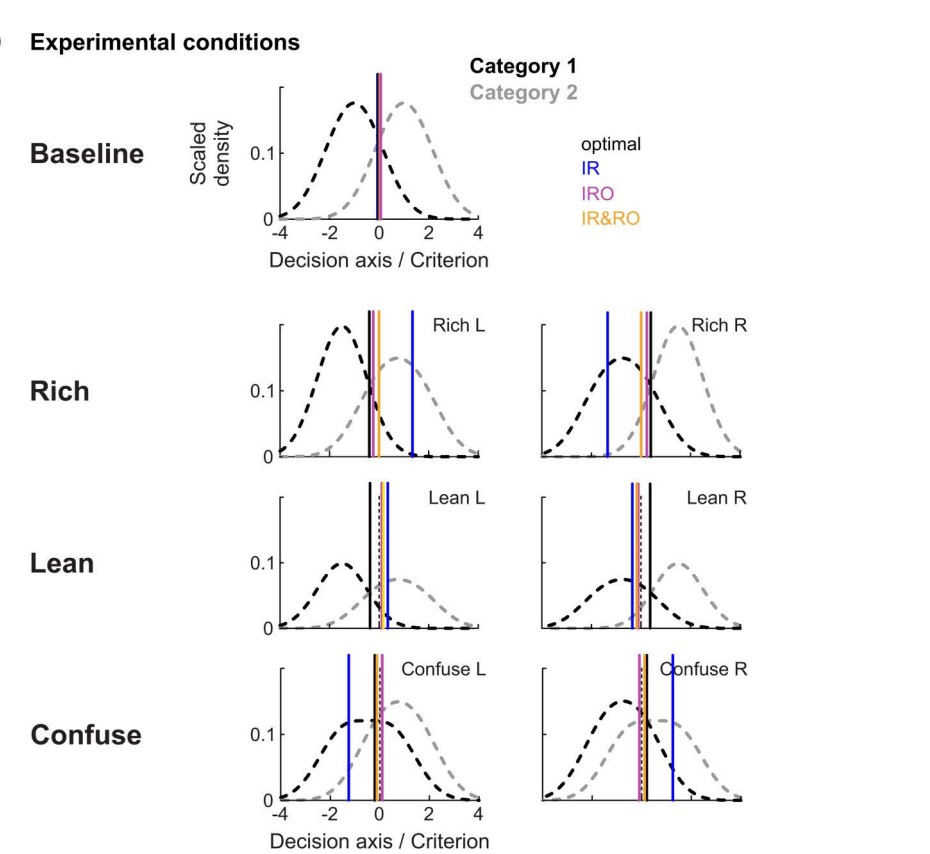

**Fig 2. Experimental design.  a.** Principle of condition design. In each condition, 3-4 out of a set of 5 stimuli were chosen (here, 3). Each stimulus (S) i was assigned to a response category (C) j and had its unique probability of presentation $P(S_j)$ and its unique reward probability $P(Rew|S_i)$ (rewards were given only when the response was correct). This example refers to the construction of condition "Lean L". Solid lines in the left and middle bottom panels represent stimulus distributions (as in Fig 1A), bold dashed lines in the bottom right panel represent 'decision distributions', i.e., the distributions for each of the two categories (C) $j \in \{1;2\}$ scaled by presentation and reward probability, i.e., for category 1, $p(x|C1)*P(C1)*P(Rew|C1,R1)$. Note that the x-value at the intersection of the two decision distributions equals the optimal (reward-maximizing) criterion (see Methods, section "Criterion setting according to optimal account"). **b.** Steady-state criterion predictions of the three criterion-setting models and a reward-maximizing account for all experimental conditions. Bold dashed lines in each panel represent the decision distributions. Solid vertical lines denote the steady-state criterion predictions of the three models and a reward-maximizing account. The parameters used for this example are $\gamma = 0.99$, $\delta = \upsilon = 0.04$. See Table 3 and S1 Fig for more details on each condition.

one stimulus that is located on the "wrong" side of the category boundary (see Table 1). Additionally, subjects underwent repeated testing in a baseline condition which serves as a neutral reference point but does not allow differentiation between the models.

We subjected rats (N=4) to a single-interval forced choice (SIFC) auditory discrimination task with multiple stimuli. Rats performed in an operant chamber with three response ports located on the side wall (Fig 3A). The trial structure is shown in Fig 3B. Trials were initiated by poking continuously for 400 ms into the center port which triggered presentation of a 100-ms auditory stimulus (chords composed of 11 pure tones of differing frequency; see Methods). Animals had to maintain poking in the center port until 25 ms after stimulus offset. Then, rats indicated their choice by entering either of the two lateral choice ports (right: R1, left: R2). If rats withdrew during initialization or stimulus presentation, the trial was aborted and excluded from analysis (see Methods for details). The stimuli were assigned to either of two mutually exclusive categories. The two categories were labeled as 'low-frequency' (Category 1, C1) and 'high-frequency' (Category 2, C2). Poking into the right side port (R1) was considered correct following presentation of a Category 1 stimulus, and accordingly for Category 2 stimuli. Correct responses were consistently rewarded in conditions Baseline, Rich, and Confuse, and probabilistically in condition Lean (see Table 1). Each session encompassed around 500 trials. Each experimental condition was typically maintained for 10 sessions. The sequence of conditions was balanced across subjects, with the exception at that the L and R versions of each condition were executed in succession.

## Experimental conditions elicit consistent response biases

Fig 3C shows the fraction of R2 responses (P(R2)) and rewarded trials (P(Rew)) per session for each of the four subjects. As per design, the Lean conditions yielded roughly half as many rewards per session as the Rich conditions (P(Rew)=0.44 vs. P(Rew)=0.87, paired t-test, p<0.001), with Confuse conditions in between (P(Rew)=0.68, significantly different from both Lean and Rich, paired t-test, p<0.01). Hit rates and false alarms (calculated for S1 and S5 respectively) increased only mildly over many months of testing (Fig 3D; increase over 80 sessions estimated through linear regression of accuracy from 0.04 to 0.11, median 0.062).

We first compared the steady-state response bias across experimental conditions. Since stimulus probabilities changed across conditions, P(R2) as shown in Fig 3C is not a suitable measure to perform this comparison, as it confounds response bias and stimulus presentation probabilities. Therefore, we used linear regression to build an SDT-based one-criterion-per-session (OCPS) model which describe performance as resulting from a session-specific criterion and three to five different stimulus distributions which were fixed across all sessions (see Methods and [51,68]). We defined the steady-state criterion as the mean criterion over the last three sessions of each condition for each animal relative to the criterion location in the baseline condition. Per definition, the value of criterion is independent of stimulus presentation probabilities and thus serves as a pure index of response bias [14]. As Fig 3E shows, rats consistently shifted their criteria towards negative values (implying a preference for rightward choices, i.e., R1) in conditions Rich L and Lean L whereas towards more positive values in conditions Rich R and Lean R (implying a preference for leftward choices, i.e., R2). Qualitatively, the shift was of comparable magnitude in the rich and lean conditions (0.52 and 0.53 for Rich R and Lean R, -0.41 and -0.51 for Rich L and Lean L, respectively; neither comparison reached statistical significance: paired t-tests, p>0.35), suggesting that the different reward densities experienced by animals in these conditions (S3A Fig) had no influence on criterion placement in the steady state. In other words, criterion setting was governed by relative rewards (which stayed constant between the Rich and Lean conditions) rather than absolute rewards (which varied across the two conditions). In the Confuse conditions, animals unexpectedly shifted their criteria towards more positive values not only in the L but also the R variety relative to baseline; accordingly, criterion values in Confuse L and R were not significantly different (paired t-test, p=0.95).

## Rats maximize reinforcement in the steady state, but none of the three models correctly predicts steady-state criteria

We next asked which of the three models (IR, IRO, IR&RO, and optimal) best predicts the condition-wise steady-state criterion locations. To that end, we first fitted rats' response data with each of the trial-by-trial criterion-learning models and then used the fitted parameters to generate predictions as to where rats' criteria would converge for each model, given a specific experimental condition and the individually fitted parameters. We then plotted the predicted against observed steady-state

Table 1. Comprehensive overview over experimental conditions. Means: stimulus means on the decision axis used in Figs 2 and S1. P($S_i$): stimulus presentation probability for stimulus i. P(Rew|$S_i$,Corr): probability of reward in trials in which stimulus i from category (C) j with j ∈ {1;2} was presented and a correct response was emitted.

| Condition | | Stimulus i | | | | |
|---|---|---|---|---|---|---|
| | | 1 | 2 | 3 | 4 | 5 |
| Baseline | Means | −1.5 | −0.5 | | 0.5 | 1.5 |
| | Category | 1 | 1 | | 2 | 2 |
| | P($S_i$) | 0.25 | 0.25 | | 0.25 | 0.25 |
| | P(Rew|$S_i$,Corr) | 1 | 1 | | 1 | 1 |
| | | | | | | |
| Rich L | Means | −1.5 | | 0 | | 1.5 |
| | Category | 1 | | 2 | | 2 |
| | P($S_i$) | 0.5 | | 0.25 | | 0.25 |
| | P(Rew|$S_i$,Corr) | 1 | | 1 | | 1 |
| | | | | | | |
| Rich R | Means | −1.5 | | 0 | | 1.5 |
| | Category | 1 | | 1 | | 2 |
| | P($S_i$) | 0.25 | | 0.25 | | 0.5 |
| | P(Rew|$S_i$,Corr) | 1 | | 1 | | 1 |
| | | | | | | |
| Lean L | Means | −1.5 | | 0 | | 1.5 |
| | Category | 1 | | 2 | | 2 |
| | P($S_i$) | 0.25 | | 0.5 | | 0.25 |
| | P(Rew|$S_i$,Corr) | 1 | | 0.25 | | 0.5 |
| | | | | | | |
| Lean R | Means | −1.5 | | 0 | | 1.5 |
| | Category | 1 | | 1 | | 2 |
| | P($S_i$) | 0.25 | | 0.5 | | 0.25 |
| | P(Rew|$S_i$,Corr) | 0.5 | | 0.25 | | 1 |
| | | | | | | |
| Confuse L | Means | −1.5 | | 0 | 0.5 | 1.5 |
| | Category | 1 | | 2 | 1 | 2 |
| | P($S_i$) | 0.25 | | 0.25 | 0.25 | 0.25 |
| | P(Rew|$S_i$,Corr) | 1 | | 1 | 1 | 1 |
| | | | | | | |
| Confuse R | Means | −1.5 | −0.5 | 0 | | 1.5 |
| | Category | 1 | 2 | 1 | | 2 |
| | P($S_i$) | 0.25 | 0.25 | 0.25 | | 0.25 |
| | P(Rew|$S_i$,Corr) | 1 | 1 | 1 | | 1 |

criteria (the latter obtained by means of the OCPS model, as above). The results are shown in Fig 4A. While the IR and IRO model predictions were only weakly (and in the case of IR, negatively) correlated with the observed steady-state criteria (IR: r=-0.56, $r^2$=0.31, IRO: r=0.20, $r^2$=0.04), both the optimal account (r=0.85, $r^2$=0.72) and the IR&RO (r=0.82, $r^2$=0.68) model provided comparably good predictions (no significant difference between the two correlations, p=0.71).

To examine whether criterion shifts were adaptive in terms of reward maximization, we compared the absolute distance of the criterion values in the first three and the last three (i.e., steady-state) sessions of each condition, relative to the

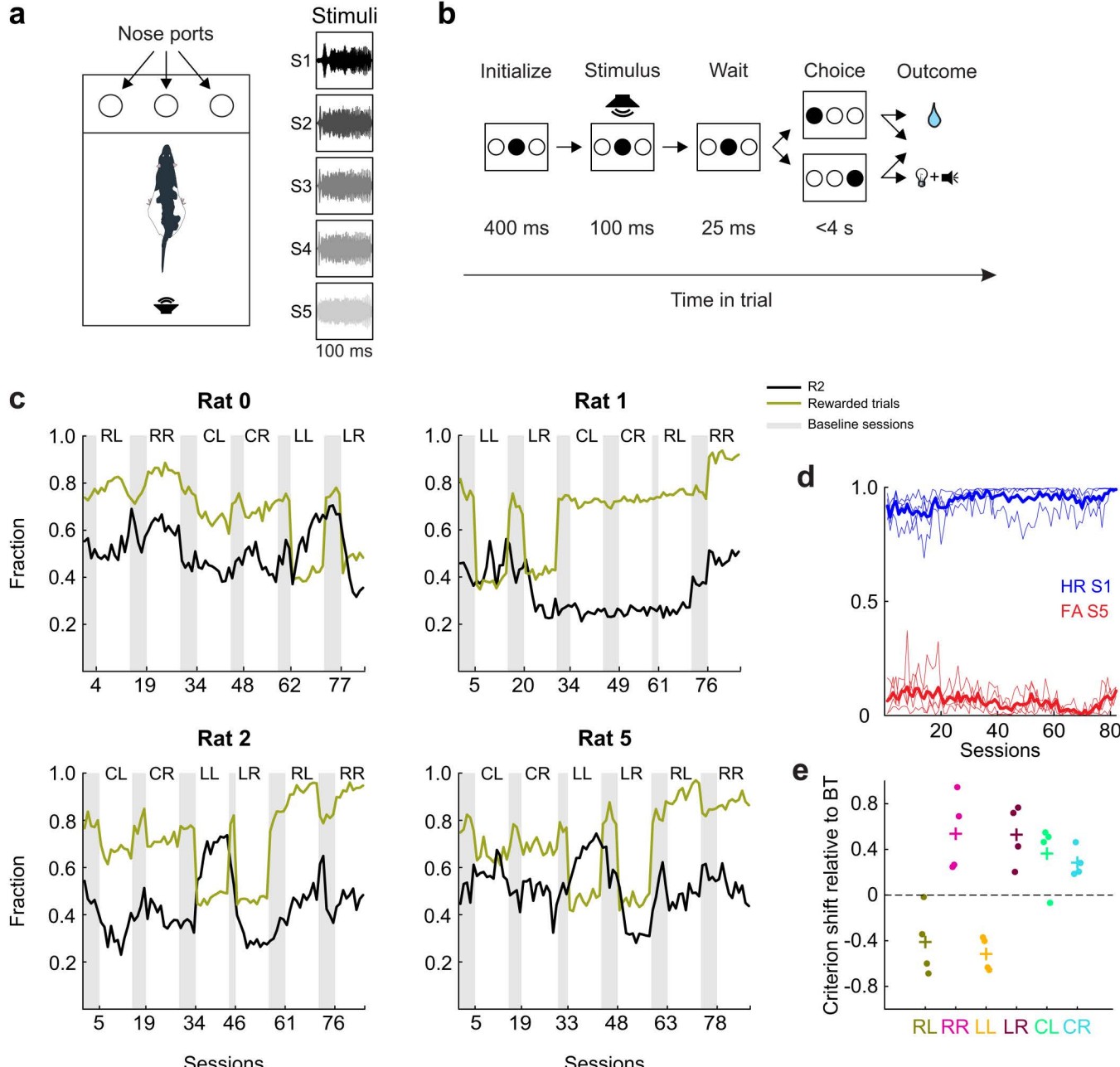

**Fig 3. Auditory single-interval forced-choice (SIFC) task and behavioral results.** **a.** Schematic drawing of the operant chamber with three conical nose ports and five representative sound waveforms used as stimuli (S1 through S5). **b.** Schematic outline of the task epochs and possible outcomes. Each rectangle represents the wall with the three nose ports (circles), filled circles represents ports which the subject is visiting in each epoch. **c.** Response bias (fraction R2, i.e., leftward responses, black) and reward density (fraction of rewarded trials, green) across all experimental conditions for all four subjects. Each panel shows results for one subject, data points represent individual sessions. Conditions are denoted by their respective initials: CL & CR for Confuse Left and Confuse Right; LL & LR for Lean Left and Lean Right, and RL & RR for Rich Left and Rich Right, respectively. The gray shaded areas highlight baseline sessions. **d.** Development of hit rate (HR, blue) for S1 and false alarm rate (FA, red) for S5 over the course of behavioral testing. Each individual line represents data from a single subject, thick lines represent the means over subjects. **e.** Steady-state criteria observed in the experimental conditions relative to criteria observed in the initial baseline sessions. Points represent the mean steady-state criterion values from the last 3 sessions of each condition for each animal, crosses represent means over the different animals. Observed session-by-session criteria were calculated using the one-criterion-per-session model (OCPS; see Methods for details).

subject-specific reward-maximizing criterion value (Fig 4B). Indeed, rats generally shifted their criteria towards values closer to the theoretical optimum by on average 0.35 (Rich, p = 0.02), 0.63 (Lean, p = 0.01), and 0.11 (Confuse, p = 0.14, all paired t-tests) units on the decision axis, thereby overall increasing the fraction of obtained rewards to ~97% of the maximally attainable reinforcements (considering imperfect stimulus discriminability).

In summary, all experimental conditions induced response biases whose directions were fully consistent across subjects. In the Lean and Rich conditions, rats shifted their criteria into opposite directions in L and R versions (moving towards the optimal criterion location), whereas in Confuse conditions rats consistently shifted their criteria towards more positive values in both L and R versions. The latter is likely an undesired artifact of experimental design: the Confuse conditions required extremely tight pre-experimental subject-dependent stimulus selection which post-hoc analysis showed to be not as intended (see S3B and S4 Figs, which shows that a) experimentally obtained decision distributions of the

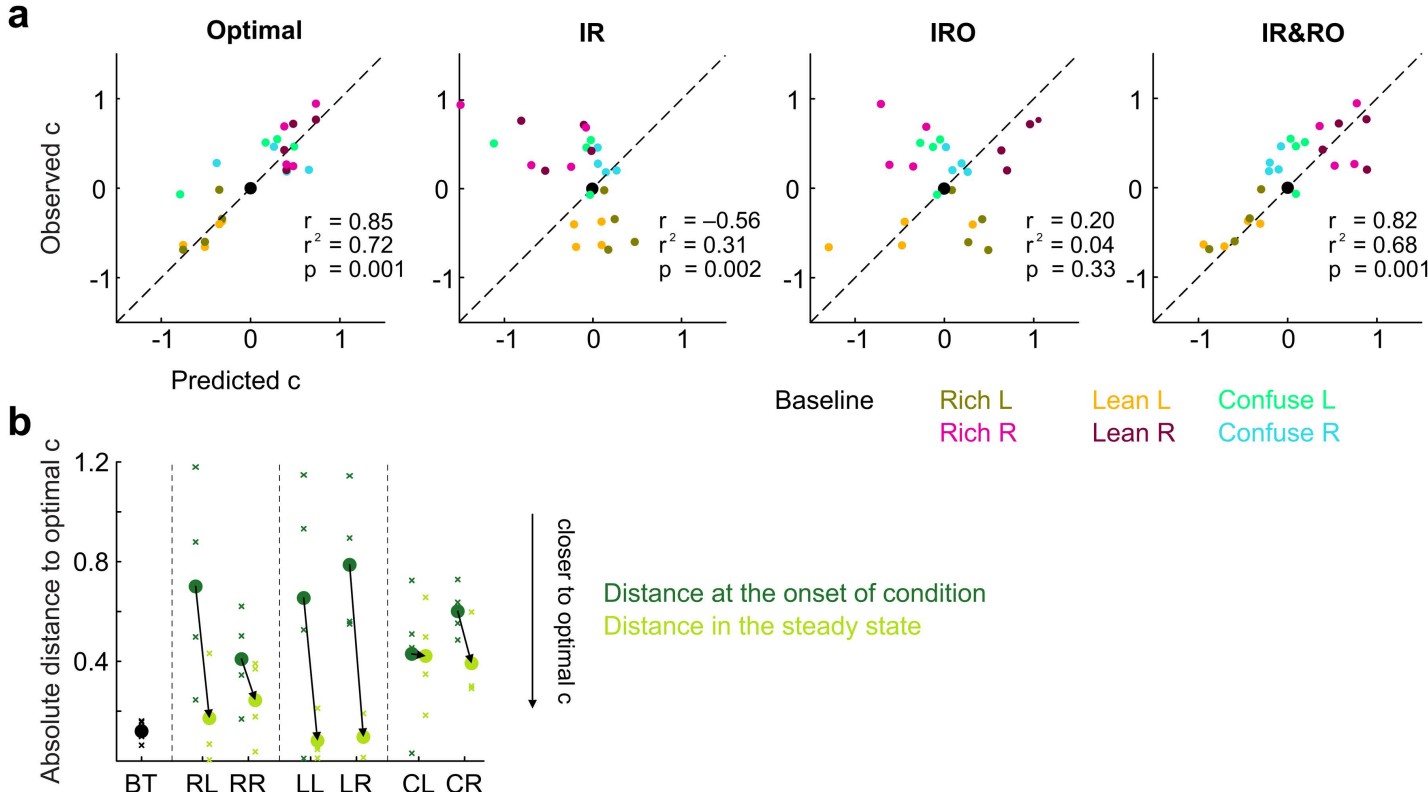

**Fig 4. Correlation of predicted and observed criterion locations in the steady states of the experimental conditions. a.** Predicted vs. experimentally observed criterion locations for the reward maximization ("optimal") account, as well as the IR, IRO, and IR&RO models. Individual data points represent a specific pair of predicted and observed mean criteria for a specific animal in a specific condition. Observed criterion locations were computed using the OCPS model (see Methods) whereas predicted criteria were obtained by solving the steady-state criterion equations (see Methods) through numerical optimization, using the fitted parameters for each rat. If predictions were perfect, data points would fall along the main diagonal (dashed line). Conditions are color-coded. $r^2$, r and p-values of the correlations are given for each model. **b.** Criterion shift from the onset (average from first three sessions, in dark green) compared to the steady states of the experimental conditions (average from last three sessions, light green), relative to the criterion that would maximize reward in the respective condition. Crosses represent individual animals; points represent means over all four animals. All criteria were normalized to baseline criterion values prior to plotting.

Confuse L and Confuse R were almost indistinguishable, and b) that predicted steady-state criterion values are all on the positive side, i.e., > 0). We will take up this matter again in the Discussion.

Notably, response bias neither conformed to the simple IR nor the IRO models, but it aligned with the predictions of the IR&RO model. We next set out to investigate the trial-by-trial performance of these models to delve deeper into the effects of rewards and reward omissions on responding.

**Simple integration of either rewards or reward omissions is insufficient to explain behavior at a trial-by-trial basis**

We have so far only looked at steady-state predictions of the learning models. One particular strength of such mechanistic models is their ability to fit choice data on a trial-by-trial basis. We fitted each model to the whole sequence of data for each animal and additionally generated 1000 simulations on a stimulus sequence obtained from a within-condition trial shuffling with the fitted parameters per animal (see Methods). Fig 5A and 5B show the results of the trial-by-trial fits of the IR, IRO and IR&RO models to the data for all rats. A model's performance should be gauged by comparing both fits and simulations to the raw data.

A first visual inspection of the IR fits and simulations confirms the results from the steady-state predictions: The IR model generally fails at qualitatively recovering the criterion shifts. We quantitatively compared the abilities of all models to fit the whole dataset through the Bayesian Information Criterion (BIC), a measure that takes into account not only the goodness of fit but also the number of free parameters in each model (differences between models larger than 6 are considered strong evidence in favor of the model featuring the lower BIC value, see Methods). The IR model exhibited drastically higher BIC values than the other two models (mean BICs for IR, IRO, and IR&RO were 24110, 23337, 23202, respectively; BIC values were all significantly different from each other, $p < 10^{-4}$, paired t-tests). As for models IRO and IR&RO, although visual inspection suggests they are better at fitting the data, the simulations show that these models are also unable to recover the learning trajectories in many of the conditions (most prominently in Rich L and Rich R).

Interestingly, both the IRO and IR&RO model fits featured negative learning rates υ for all four animals (range -0.01 to -0.03; Fig 5B). This is at odds as to how these models have been designed to function since it implies that animals shift their criteria as to emit unrewarded responses more rather than less often, and thus compromises the models' interpretability. The main reason for obtaining the negative learning rates is likely the presence of autocorrelation in the response data: as per design, we induced a response bias, which implies that the probability of the preferred response to occur after any other response is > 0.5 in the steady state (for extended treatment, see [69].

**Rewards do affect subsequent behavior while reward omissions have no discernible effect**

Due to the inability of all trial-by-trial models to adequately recover response patterns, and the counterintuitive negative υ learning parameters returned by the IRO and IR&RO models, we took one step back and analyzed the influence of rewards and reward omissions on subsequent choices from a different perspective. To that end, we regressed the response on trial t as a function of the current stimulus as well as outcomes (rewards and reward omissions) in the preceding trials (t-1, t-2, t-3 and t-4) [46,73,74]. One would expect rewarded responses to have positive weights (assuming they lead animals towards repeating rewarded responses) and reward omissions to have negative weights (assuming they influence animals towards choosing less often the unrewarded response). This analysis showed that responses were indeed influenced by past rewards whereas the influence of reward omissions was practically nonexistent. On average, regression coefficients for responses following reward omissions where four- or five-fold smaller relative to those of rewards (e.g., 0.43 vs. −0.08 for trial t-1 and 0.19 vs. -0.06 for trial t-2; see Fig 5C). Taken together, the small regression weights, along with the consistently negative υ learning rates of the fits returned by the IRO and IR&RO models for all subjects, suggest that reward omissions did not influence the rats' responses in our task, unlike rewards. We therefore moved on to reconsider only the reward-learning model IR.

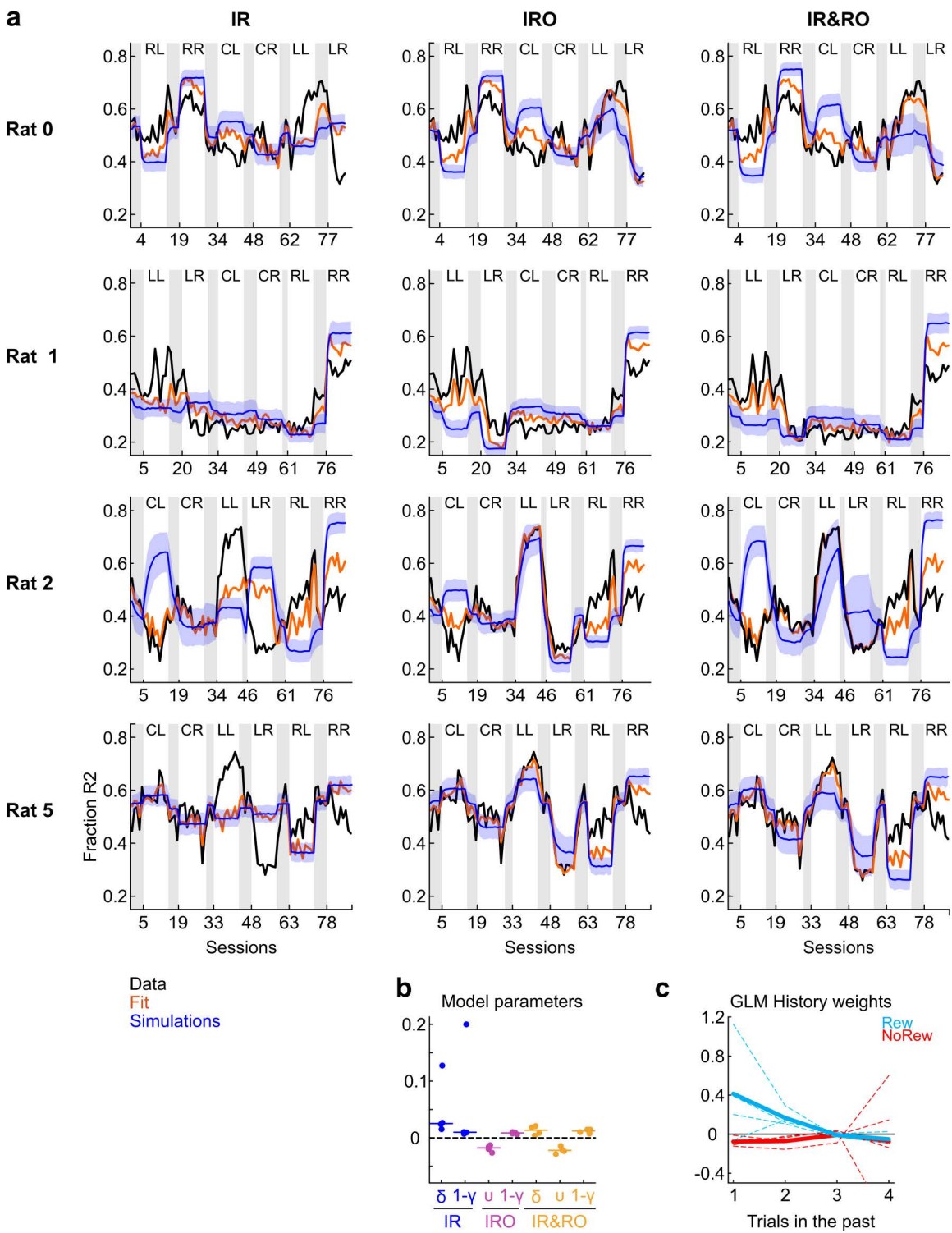

**Fig 5. Trial-by-trial fits of the IR, IRO and IR&RO models to the experimental data. a.** The fraction of leftward responses in each session, P(R2), is plotted for each individual animal across the different experimental conditions, similar as in Fig 3C. Orange lines are model fits. Blue lines are averages over 1000 simulations; blue shaded areas represent ±1 SD. **b.** Distributions of fitted parameters of the different models. Each data point pertains an individual subject. **c.** Regression weights for rewards and reward omissions. The regression weights indicate the influence of both types of outcomes (blue for rewards and red for reward omissions) for trials t-1, t-2, t-3 and t-4. Dashed lines represent individual rats and thick lines means across the four subjects. See Methods for details of the GLM fit.

## Incorporating stimulus-specific learning rates into the IR model is key to explaining learning trajectories

Confronted with the finding that decisions were indeed influenced by past rewards but hardly by reward omissions, but also with the IR model's inability to both fit and adequately reproduce learning trajectories in simulations, we were forced to consider additional mechanisms underlying learning from rewards not captured in the original IR model.

We will first reconsider the learning rate δ. In the IR model, δ is a single fixed value. We equipped all our initial models with fixed learning rates for reasons of parsimony, but there is reason to believe that this is an oversimplification. In classical learning theory, the learning rate is generally thought to rely on prediction errors, i.e., the size of the difference between expected and obtained outcomes [75]; (for example, appetitive discrimination learning proceeds faster with larger rewards than smaller ones [76,77]. In PDM tasks, trials featuring difficult stimuli (i.e., stimuli that lie close to the category boundary) elicit low reward expectations, leading on average to larger prediction errors following rewards and therefore larger update steps [49,66]. Generally speaking, learning is larger after non-predicted outcomes, and this can be adaptive in volatile environments [78]. Accordingly, to enable the IR model to capture the relation between stimulus uncertainty and update size, we implemented stimulus-dependent learning by fitting a model with one learning rate per stimulus (yielding five learning rates, compared to only one in the original IR model). We coined this extended model *Integrate Rewards with Stimulus-specific Learning Rates* (IR-SLR).

When fitting the datasets with this model, two results became apparent. First, the quality of the fits increased abruptly for all animals (see Fig 6A and 6B for an example animal, S6A and S6B Fig for the others), as becomes apparent when comparing their BIC values (average difference IR and IR-SLR: 1547, range: 810–3115, all by far favoring IR-SLR; also see Fig 6D). Relatedly, forward simulations of this model also qualitatively reproduced the observed behavior in most conditions (Figs 6A and S6B, blue shading). Second, as previously observed [49,66], there was a negative correlation between learning rates and discrimination difficulty (the distance of the stimulus means to the category boundary), i.e., rats shifted their criteria by a larger amount following rewards in trials where the discriminative stimulus is close to the category boundary relative to stimuli further away from it (r = -0.91, p < 10$^{-7}$ between absolute stimulus means and normalized learning rates; Fig 6C). These results suggest that criterion adjustment is not the same in all rewarded trials, but instead depends on the degree to which a reward is expected based on the discriminative stimulus in that trial.

## Criterion in the steady state is determined by relative rather than absolute reward differences

While the introduction of stimulus-specific learning rates to the IR model dramatically improves model fits, this model still consistently underestimates the steady-state response bias in the Lean relative to the Rich conditions (Figs 6A and S6B, see esp. rats 2 and 5, average undershoot 0.12, range 0.06 to 0.19), which is at odds with our previous observation that animals reached similar steady-state criteria in Lean and Rich conditions (Fig 3E) and which suggested that overall reward density did not seem to affect rats' steady-state behavior.

Importantly, all reward-learning criterion-setting models we discussed so far predict different criteria in the steady state of Rich and Lean conditions (S1 Fig). For the criterion that the IR model converges to, it holds that $c = \delta/(1-\gamma) * (\mathbb{E}[Rew_{R1}] - \mathbb{E}[Rew_{R2}])$ (see derivation in the Methods section), so the criterion position depends on the absolute difference in reinforcement obtained for R1 and R2. This difference scales with the overall reward density: changing the overall reward density (while keeping the reward ratio between R1 and R2 the same) will also change the absolute difference in reinforcement. For example, when the reward rates for both categories are doubled from one condition to another one, the absolute difference in reinforcement also doubles. Hence, the predicted steady-state criterion for the Rich conditions is higher than the one for the Lean conditions, which have a comparatively lower reward density. However, as reported above, we found the steady-state criteria to be very similar in the Lean and in the Rich condition in our experiment. Accordingly, the IR-SLR requires modification to be consistent with this finding.

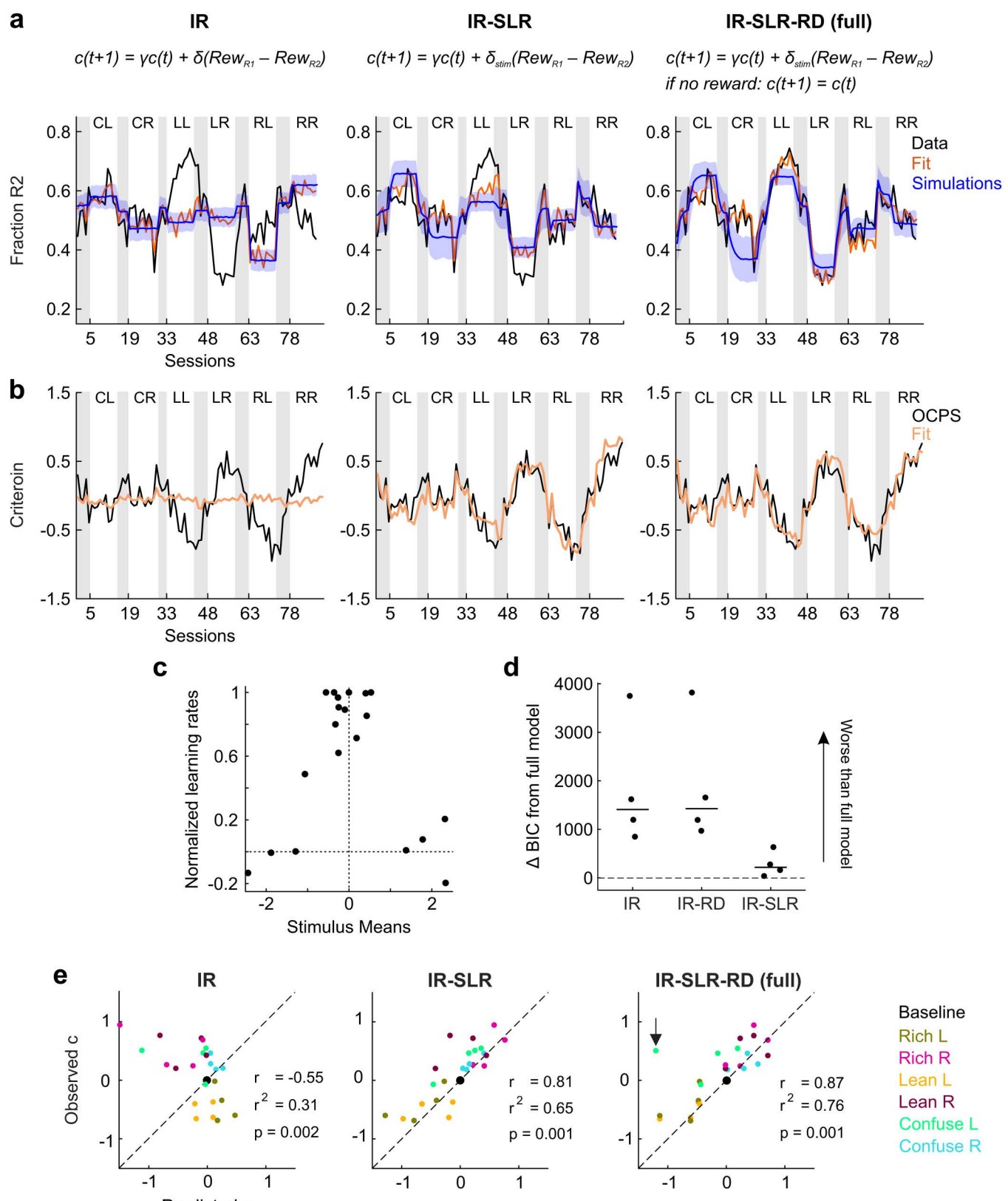

**Fig 6. Session-by-session and steady-state performance of the IR, IR-SLR and IR-SLR-RD models. a.** Fit and simulation results for the two new model versions (IR-SLR and IR-SLR-RD, IR replotted for comparison purposes) for an example animal (rat 5). Format as in Fig 5A. IR-SLR differs from

the original IR model only in the number of learning rates δ (IR: 1, IR-SLR: 5). IR-SLR-RD additionally differs from both other models in that the criterion does not decay towards 0 on unrewarded trials, i.e., when $Rew_{R1} = Rew_{R2} = 0$. **b.** Same as in a, but plotting session-by-session criteria. **c.** Stimulus-specific learning rates returned by the IR-SLR model as a function of the fitted stimulus means. For comparison purposes, all values were normalized to the overall highest value. **d.** Model comparison through the Bayesian Information Criterion (BIC). In this panel, relative values are shown, i.e., the BIC of the IR-SLR-RD (i.e., "full") model was subtracted from that of the other models, so that positive values are indicative of worse fits than the full model. **e.** Same as in Fig 4A, but comparing steady-state criterion prediction performance of models IR-SLR and IR-SLR-RD with the basic IR model. In the IR-SLR-RD plot, r, $r^2$, and p-values exclude outlier (highlighted by a black arrow); including the outlier, the resulting values are r = 0.71 and $r^2 = 0.51$.

However, this feature need not be implemented by adding an additional reward density-dependent parameter to control learning rates, but we can instead simply restrict the pull-back uniquely to rewarded trials. In doing so, it can be shown that the steady state to which the model converges to is $c = \delta/(1-\gamma) * (\mathbb{E}[Rew_{R1}] - \mathbb{E}[Rew_{R2}])/ (\mathbb{E}[Rew_{R1}] + \mathbb{E}[Rew_{R2}])$ (see Methods section). So, the steady state is not anymore determined by the difference between the expected rewards obtained from the two categories, but by the difference in the relative reward rates, which also means that the model becomes insensitive to reward density. Importantly, this minor modification brings the model a step closer to the predictions of the matching law [70], which predicts that the relative response proportion in a two-choice situation is a function of the relative proportion of rewards obtained from the two choices (see Discussion).

The new version of the model, dubbed IR-SLR-RD (to make direct reference to the fact that steady-state behavior is controlled by relative differences rather than absolute ones), indeed shows clearly improved fits and simulations for the Lean conditions in all rats (Fig 6A and 6B for an example animal, see S6D Fig for data from other subjects). As expected, the IR-SLR-RD model outperformed all other models with respect to BIC (Fig 6D) by at least 44 (on average, ΔBIC = 279 when comparing IR-SLR-RD against IL-SLR, and ΔBIC = 1825 when comparing IR-SLR-RD against the original IR model). This improvement resulted in part from a better fit in the Lean conditions; inclusion of RD modification reduced undershoot for Lean L and Lean R (IR-SLR: 0.12, s.a.) to 0.06 and 0.03, resp. (IR-SLR-RD).

Lastly, we examined the predictions of the IR-SLR and IR-SLR-RD models for the steady state and found that these predictions were indeed able to both qualitatively and quantitatively capture the steady state of most conditions in all animals (IR-SLR $r^2 = 0.65$, IR-SLR-RD $r^2 = 0.76$ (without outlier)), indicating that these modifications together were key to fit rats' performance particularly when compared with the basic IR model (IR $r^2 = 0.31$) (Fig 6E; see S6A-H Fig for fits of all models to all subjects).

### The IR-SLR-RD model generalizes to pigeons performing a visual PDM task with the same experimental conditions

Quantitative research on adaptive perceptual decision-making is mostly conducted with rats, mice, and humans (see references in Introduction). However, learning per se is of course not restricted to mammals but present in all studied vertebrates [79], and general principles of learning appear to be highly conserved across animals generally [80,81]. Therefore, to assess the generality of our results, we analyzed data from an additional experiment in which the same battery of experimental conditions was run with four pigeons as subjects. The birds performed a structurally similar perceptual choice task which however featured visual (shades of gray differing in luminance) rather than auditory stimuli and food rather than water as reinforcer (Fig 7A and 7B; see Methods for further details).

All major results from the rat experiments were replicated with the pigeon data. First, all animals adapted more profitable (=closer to optimal) criterion values within each experimental condition (on average by 0.17 (Rich), 0.51 (Lean), and 0.35 (Confuse) units on the decision axis; Fig 7C). Second, again neither IR nor IRO models were good predictors of steady-state criteria (Fig 7D), while the combined IR&RO and optimal accounts were better in quantitative terms ($r^2$) but still far from satisfactory (IR&RO $r^2 = 0.15$, optimal $r^2 = 0.31$). Third, the υ parameter was consistently negative for all considered models (range -0.02 to -0.1; Fig 7F), suggesting that reward omissions are not a major determinant in adaptive criterion setting. Fourth, augmenting the IR model with an SLR modification improved the fit in most conditions, although

the model was still underestimating steady-state criteria in the Lean conditions (Fig 7E). Fifth, restricting leaky integration to rewarded trials improved the performance in the Lean conditions in the same way as for rats (Fig 7E) and generally provided both excellent fits and forward simulations. Finally, inspection of the BIC values shows that again the IR-SLR-RD model fared best (on average, $\Delta BIC = 115$ in favor of the full model; Fig 7G). Furthermore, this model also exhibited the largest correlation between steady-state predictions and the experimentally obtained criteria (IR-SLR(red) $r^2 = 0.52$, IR-SLR(red)-RD $r^2 = 0.59$, Fig 7H).

## Discussion

We set out to describe adaptive perceptual decision-making under a broad variety of stimulus-response-outcome manipulations. To that end, we initially considered three different SDT-based criterion learning models and examined their ability to fit trial-by-trial response data in various experimental conditions, generate qualitatively similar data in forward simulations, as well as predict steady-state criterion values. We found that neither of the three models was able to account for the patterns in the data, suggesting that these models are missing essential components of adaptive behavior. After confirmation that indeed past rewards rather than reward omissions influence choices (above and beyond the discriminative stimuli), we introduced two modifications to the model which have been suggested as key determinants for both trial-by-trial learning and steady-state performance before, but that have so far not been considered in conjunction. These manipulations were first, making criterion updating dependent on stimulus discriminability, and second, making the steady-state of the models dependent on differences in relative rather than absolute reward rates across the two categories. These modifications increased the performance of the reward integration model with respect to data fits, simulations, and predictions of steady-state criteria. Moreover, the importance and generality of these modifications are supported by our finding that they also proffered similar increases in performance in a second dataset collected with pigeons, which were tested with stimuli from a different modality (vision instead of audition) and obtained a different type of reinforcer (food pellets instead of water) but otherwise were subjected to the same experimental manipulations. Interestingly, both features are advantageous in certain scenarios: making learning dependent on uncertainty increases reward rate in volatile environments [82], and matching response ratios to relative reward ratios in the steady state can approximate a reward maximization strategy [83].

### Animals maximize rewards

Both rats' and pigeons' steady-state criteria were close to optimal – i.e., reward-maximizing – values. Optimality is frequently taken to be a useful benchmark for gauging performance [8,84–88], and indeed both human and animal performance approaches optimality in a variety of settings (perceptual: [89]; value-based: [90]; environmental volatility: [82]; cross-modal integration: [91], timing: [92]). Importantly, "optimal" does not imply perfect, because animals are usually uncertain about various aspects of the experimental conditions that they find themselves in (e.g., uncertainty about a stimulus as posited in SDT, or uncertainty about the estimation of time or stimulus probabilities, see, e.g., [93]. Optimality is, hence, defined relative to specific assumptions about what information is or is not available to the animal. With respect to perceptual decision-making under reward uncertainty, we previously reported near-optimal criterion setting (in the steady state) in pigeons and proposed a model how this could be achieved [51]; also see [54].

On the other hand, performance has frequently been shown to be clearly suboptimal. For example, Berkay and colleagues [94] found that mice and rats fail to incorporate exogenous noise into their timing judgments, and we have previously reported that rats' steady-state criteria consistently were less extreme than required for optimality [69,95]. We also found that within the same task, and with the same subjects, a seemingly minor variation of reinforcement contingencies produces quasi-optimal criterion setting in one experiment [51] but is clearly suboptimal in another [68]. This last experiment was set up specifically to demonstrate that animals can be misled in a way that their criteria correlate negatively with the optimal criteria in an appropriately chosen set of conditions. Therefore, a model that assumes that animals optimize rewards under all conditions is obviously false, even if it is a good first approximation in many cases.

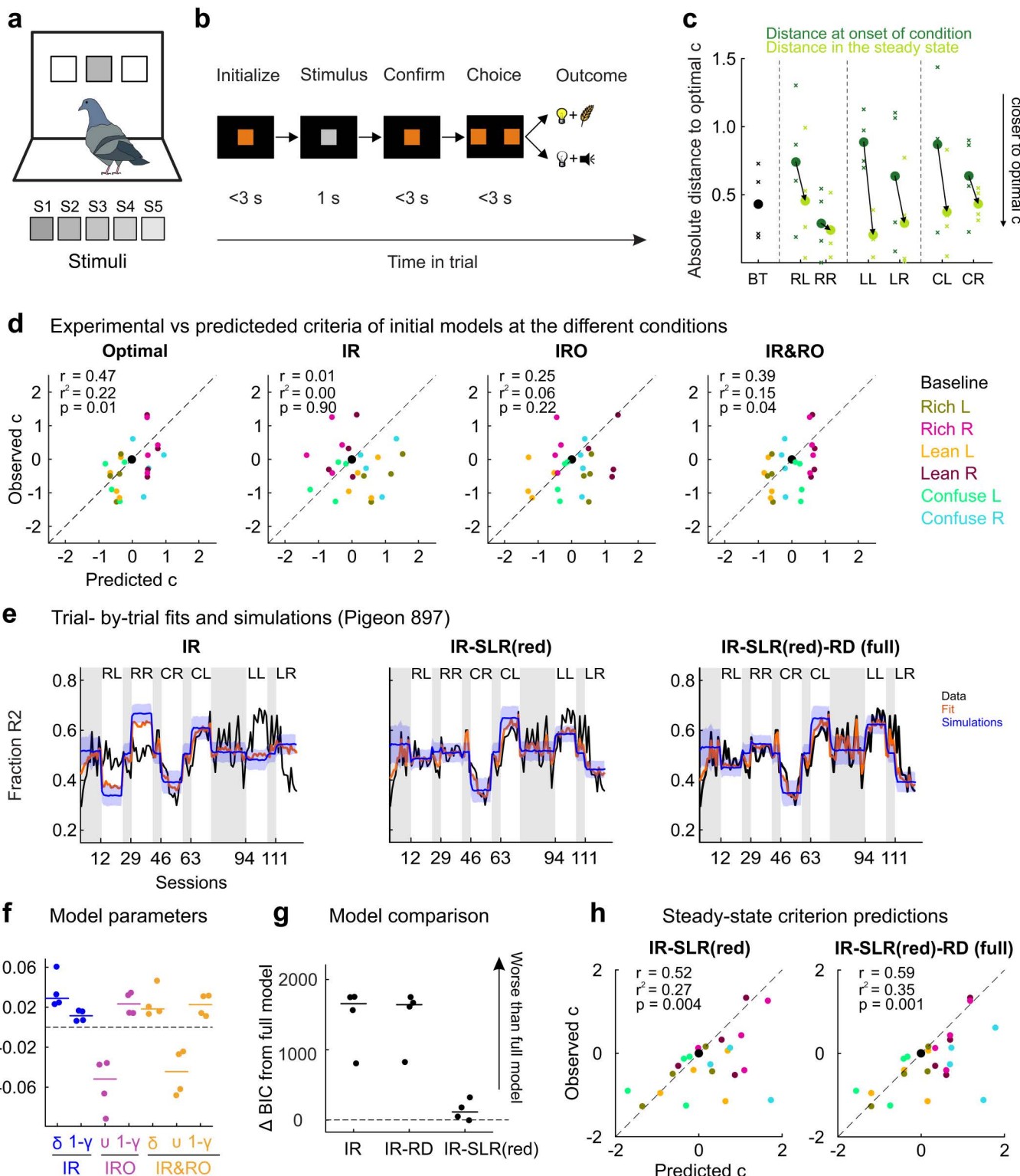

**Fig 7. Experimental setup and results from a second dataset from pigeons performing a visual task. a.** Pigeons were tested in operant chambers. One side of the chamber featured a touch screen on which three horizontally aligned rectangular areas were designated as "pecking keys". On these keys, visual stimuli were displayed, and key pecks within these areas were counted as responses. In each trial, one out of five possible

discriminative stimuli (shades of gray, numbered S1 through S5) was presented on the center key. **b.** Timeline of an example trial. Each trial started with orange illumination of the center key. Following a single peck at this key, the discriminative stimulus was presented for one second. Thereafter, the center key again turned orange, and the pigeon had to emit a single peck at the key to turn it off and illuminate the two side (choice) keys. Pecking at a choice key was followed either by food tray illumination and food delivery, or by a negative feedback sound and the turning off of the house light. See Methods for further details. **c.** Criterion shifts from condition onsets (dark green) to steady states (light green) relative to the reward-maximizing criterion for all subjects and experimental conditions. Format as in Fig 4B. **d.** Comparison of the initial criterion-setting models (IR, IRO, IR&RO and optimal account) in their ability to predict steady-state criteria for all subjects and conditions. **e.** Fits and simulation results of the IR, IR-SLR and IR-SLR-RD models to response data from an example pigeon (subject 897). Format as in Fig 5A. **f.** Model parameters returned by each of the initially considered models. **g.** Model comparison of all the models featuring reward learning. **h.** Steady-state criterion prediction performance of the IR-SLR and IR-SLR-RD models with the IR model in the pigeon dataset.

One particularly well-studied reason for the observed suboptimality is criterion variance [24,25,38,96,97]), which all of our models explain not as a random process but as a systematic adjustment of the criterion which serves to produce rewarded responses more frequently. Importantly, none of our models is inherently optimal, as we have shown before [68].

The many reasons for suboptimal performance have been reviewed comprehensively, and the general usefulness of optimality as a benchmark for performance is intensely debated, but so far this benchmark has not been abandoned [85,98–101]; also see discussion in [68]). At present, the upshot seems to be that animals perform near-optimally in many scenarios (including the experimental conditions in this study) and suboptimal performance can be informative with respect to the assumptions made when defining "optimal performance" in the first place [102]. Keeping optimality as a benchmark seems to be useful mainly because it serves as an initial hypothesis at the computational level before more realistic mechanistic models can be developed at the algorithmic and implementational levels [85].

## Rewards and not reward omissions determine adaptive behavior in our task

We found that fits of all models that incorporated criterion updates after reward omissions (IRO, IR&RO and these same models with the SLR and RD modifications) consistently produced negative learning rates. Under our model architecture and under a lose-switch policy, negative learning rates should not occur as they imply animals shift their criteria as to repeat the last unsuccessful response. We have demonstrated elsewhere that negative parameters are unable to qualitatively reproduce experimental response data in forward simulations [69]. Here, we additionally used logistic regression to examine the impact of reward omissions on responding, and confirmed that rewards rather than reward omissions consistently influenced responding in our task.

Generally, there are more behavioral states that can lead to an unrewarded trial than to a rewarded one: From not paying attention to the sensory evidence during stimulus presentation [103], to fluctuating levels of motivation [104], to exploring uncertain over exploiting known stimulus-response combinations [105], all these are more likely produce unrewarded rather than rewarded outcomes [49]. As a consequence, and given the behavioral variability associated with an unrewarded trial, the associations between stimulus-response-outcomes are weaker in trials that end up in reward omissions than those that end up being rewarded. In that sense, our results are in accordance with the literature in that animals display a high degree of response variability after an unrewarded trial which cannot always be attributed to a single learning algorithm [49]. Additionally, and specific to some of our experimental conditions, unrewarded trials provide less information than rewards about the currently effective stimulus-response-outcome contingencies. Particularly, in Lean conditions, only 50% of correct responses were rewarded, so animals received definitive feedback on the S-R-O contingency when rewarded but ambiguous feedback when the reward was omitted (because reward implies a correct response, while reward omissions inform only probabilistically on whether the response was correct or not). This particular experimental manipulation may additionally contribute to why reward omissions were not relevant in determining animal performance. Accordingly, future experiments might test the generality of this finding in other designs, e.g., where reward omissions

are more informative about the currently effective S-R-O contingency, or where reward omissions are accompanied by aversive stimuli such as foot shocks.

## Criterion adjustment depends on stimulus uncertainty and increases expected reward

We found that in both rats and pigeons, reward-learning models could capture and reproduce adaptive choice behavior only after including stimulus-specific learning rates. The fitted learning rates varied systematically as a function of the stimulus means, exhibiting an inverted-U-shaped distribution, with learning rates large for stimuli whose mean was close to the category boundary (featuring high perceptual uncertainty) and small for stimuli whose mean was further away (featuring low perceptual uncertainty; Fig 6C).

In perceptual choice tasks such as ours, easily discriminable stimuli are associated with correct responses and subsequent reward delivery more often than more difficult stimuli. According to a reward prediction error framework, rewards following easy stimuli are largely predicted, rewards following more difficult stimuli however are not. For example, rats estimate the perceptual uncertainty of a decision and use it to guide their behavior [106]. Relatedly, in monkeys performing a random-dot motion discrimination task, the dopamine neuron activity evoked by discriminative stimuli increases with motion coherence (i.e., decreases with perceptual uncertainty), and the same dopamine neurons were found to signal reward prediction errors [107]. Lak et al. [108] proposed that dopaminergic neurons signal the inverse of perceptual uncertainty, i.e., decision confidence, the degree of belief that a particular stimulus belongs to a given response category. These authors also showed that decision confidence modulates trial-by-trial learning from rewards and continues to do so even after several months of training [49,66]. Importantly, subjects in that latter study performed under constant reinforcement contingencies, so there was no need to adjust performance to varying experimental conditions. Accordingly, in their study any type of adaptation was disadvantageous since consecutive trials were independent.

We show here that in the context of an adaptive design in which S-R-O contingencies change infrequently (every one to two weeks constantly over many months of testing), adaptive criterion setting allows the subjects to harvest more rewards, as subjects' decision criteria move closer to the reward-maximizing criteria in each condition. Indeed, steady-state criterion predictions were highly similar for the optimal and the IR-SLR accounts, in stark contrast to the predictions from the simple IR model in which these predictions were orthogonal to these two accounts (for rats and pigeons respectively, optimal vs IR (r=-0.48, r=-0.19), optimal vs IR-SLR (r=0.69, r=0.58) and optimal vs IR-SLR-RD (r=0.66, r=0.71). The fact that stimulus-specific learning is beneficial in the context of an adaptive task and that it is present even when the adaptive component is lacking (as in [49]) suggests it to be an integral feature of animal decision-making which is beneficial in volatile environments, i.e., when S-R-O contingencies are not stationary.

## Adaptive behavior under reward uncertainty

In our experiments, subjects experienced not only stimulus uncertainty but also uncertainty related to reward. Within experimental conditions, reward uncertainty arises mostly due to stimulus uncertainty (because all correct responses were rewarded in all but Lean conditions). Additionally, S-R-O contingencies changed multiple times over the course of the experiment (Fig 3C). This type of outcome uncertainty is termed "volatility" or "unexpected uncertainty" (for review, see [109]) and is commonly investigated in dynamic foraging tasks in which pairs of unequal reward probabilities for two choice options change blockwise; the number of trials per block (usually 40–200) then constitutes the independent variable "volatility" (e.g., [78,110,111]).

While theoretical work suggests that optimal learning rates are larger in more volatile environments [112,113], experimental evidence is equivocal, with some studies reporting higher learning rates in more volatile environments (e.g., [82]), and other studies finding the opposite effect [111]. In our experiments, animals experienced a low-volatility environment, with conditions changing only after thousands of trials. On a normative account, one would expect stable learning rates,

which is consistent with our finding that a fixed set of stimulus-specific learning rates is sufficient to fit the choice data well (Figs 6A and 7E).

Relatedly, while we have not systematically manipulated volatility in the present study, we have previously performed fits of the IR model to data from a highly similar perceptual decision-making task in which conditions changed every 200 trials [95]. In that study, values of the learning rate parameter $\delta$ were much higher than in the present one (median $\delta = 0.11$, this study: median $\delta = 0.03$), in line with the hypothesis that animals adapt learning rates to the volatility of the environment. Accordingly, in an experiment involving both high- and low volatility conditions, adaptive learning rate parameters might be necessary to adequately fit the choice data (see, e.g., [90]).

## Steady-state biases depend on differences in relative rather than absolute rewards

Although the introduction of stimulus-specific learning rates to the IR model dramatically improved both fits and simulations in the Confuse and Rich conditions, the IR-SLR model still dramatically underestimated P(R2) in the Lean condition (Figs 6A and S6B). Prompted by our finding that steady-state criteria in the Rich and Lean conditions were of comparable magnitude (Fig 3E), despite differing in reward density (animals obtained twice as many rewards per trial in Rich compared to Lean conditions, see Fig 3C), we hypothesized that relative rather than absolute reward rates determine subjects' steady-state biases. This hypothesis is in line with a classic notion in animal learning theory – Herrnstein's matching law [70]. The matching law posits that choice allocation for two response alternatives (i.e., R1/(R1+R2)) is proportional to the relative reinforcement obtained from these alternatives (i.e., Rew1/(Rew1+Rew2)). The matching law has been confirmed in a large variety of human and animal decision-making scenarios [71,114,115]. By restricting the pull-back of the criterion $\gamma$ to rewarded trials in the IR-SLR-RD model, the steady-state criteria become directly proportional to the difference in relative rewards. This is similar but not identical to the prediction of the matching law, which is that the criterion is proportional to the difference in their logarithm. For details, we refer the reader to our theoretical work aiming to develop a trial-by-trial model which is fully congruent with the generalized matching law [72].

The RD modification makes the steady-state criterion independent from global reward rate/ reward density, consistent with our present findings and in line with the matching law. However, reward density has been found to affect decision-making in other contexts. For example, pigeons' response rates increase monotonically as a function of reward density [116], and monkeys tend to repeat successful choices in rich environments but abandon unsuccessful choices in lean environments [117]. With respect to response times, mice respond faster when the animals' current estimate of reward density was high [110], and similarly pigeons respond faster in blocks with high compared to low reward density, with stimulus discrimination performance remaining unaffected [118].

While we found no influence of reward density on choice allocation, analysis of reaction times (time intervals from stimulus onset to response registration) showed that animals tended to respond somewhat more slowly in Lean than in Rich conditions (54 ms across all animals, averaged across response directions and L/R variants; p = 0.086, paired t-test), mirroring the results of [110] and [118]. Our models do not make predictions regarding response times, but our model could be combined with sequential sampling or drift-diffusion models [119] by conceptualizing the criterion as the starting level of the accumulation process [118]

## Unexplained variance in the Confuse condition

The full model (IR-SLR-RD) provides excellent fits to the choice data for both rats and pigeons. In addition, forward simulations of the model generate similar choice patterns as those obtained experimentally in the baseline, Lean L, Lean R, Rich L and Rich R conditions. However, the model simulations produce discrepant behavior some of the Confuse conditions for some animals (S6D Fig, especially Confuse L in rat 2, and S6H Fig, pigeons 666 and 902).

The main reason for this is an unwanted consequence of the experimental design, namely the high similarity of the two Confuse conditions stimuli 2 and 4 which produced highly similar behavior despite differing S-R-O contingencies. Rats exhibited similar steady-state criteria in the two conditions (Fig 3E). Closer inspection of the decision distributions as well as the objective reward functions in S4 Fig shows that the two Confuse conditions produce highly similar decision distributions on the decision

axis, and correspondingly predict similar steady-state criterion values. For all four rats, steady-state criteria exhibit positive values and therefore lie on the same side of the category boundary (as should be for Confuse R but not for Confuse L).

Also, we cannot rule out that the design-inherent confusing S-R-O contingencies in the Confuse conditions, providing ambiguous feedback for stimulus-response combinations due to the similarity of the two central stimuli (S3 and S4 in Confuse L and S2 and S3 in Confuse R, see Fig 2) which however were assigned to different responses, might have led the subjects to "tune out" of the task. Indeed, it has been widely reported that animals leverage different response strategies in PDM tasks associated to varying levels of engagement, or task proficiency [104,120,121], and one limitation of our modeling efforts is that this source of variability is neglected.

## Limitations and future directions

Surprisingly, despite the central role of the criterion concept in SDT, there is a paucity of work on criterion setting at a trial-by-trial (but not the steady-state) level [21,25,50,63,97]; reviewed in [14]). We based our initial models (IR, IRO, and IR&RO) on the pioneering works of Kac, Dorfman and colleagues [55,57,67] and we have documented several of their limitations [68,69]; present work). Other SDT-based models of trial-by-trial criterion setting involve similar criterion updating mechanisms (e.g., [33,39,58,122]; see [64], and [123], for somewhat different approaches) but largely focus more on perceptual rather than reward-related aspects (e.g., [59,60]), which makes our approach unique. That said, Lak and colleagues [47,49,66,108]; also see [124]) have recently developed an RL-based model of adaptive perceptual decision making that could, with some modifications, be applied to our data. The comparison of RL- and SDT-based models for adaptive perceptual decision-making is an interesting avenue for future research.

While our model integrates many aspects of perceptual decision-making such as stimulus presentation probability, reward probabilities, and stimulus-category assignments, it is silent on many other important aspects.

1) The model does not explain how the task and the criterion are learned in the first place. For example, it is unclear how stimulus category representations emerge [23], how animals learn the initial stimulus-response mapping, and how they translate these representations into performance [21,121,125–127]. A more thorough investigation of task learning is not only important in itself but can also help to refine models which describe criterion setting after task acquisition, such as ours, and it could also connect adaptive criterion setting to commonly employed tasks of cognitive flexibility such as reversal of stimulus-response mappings (i.e., S1→R2, S2→R1; [128]).

2) Similarly, the model is silent on how stimulus-specific learning rates develop and how they relate to decision confidence (e.g., [106,129]). Notably, our model features fixed stimulus-specific learning rates, while Lak et al.'s [108] RL-based framework posits variable learning in each trial, with the updating depending on the current stimulus as well as dynamically changing reward expectations for each response option. The question whether learning rates are fixed or variable is intensely discussed (see, e.g., [109]).

3) The model assumes that animals are constantly engaged in the task and do not exhibit fluctuating and criterion-independent biases nor lapses. While we contend that lapses occurred only infrequently in our highly experienced subjects (Figs 3D and S7B), we cannot rule out drifts of response bias and periods of task disengagement such as described by Ashwood et al. [120]. That said, assuming that our animals are working in the task-engaged state most of the time (as in the data set used by Ashwood and colleagues), fluctuating biases and random responding would primarily decrease and offset our estimates of stimulus means, but not affect our two main conclusions about the role of stimulus uncertainty and the importance of relative reward rates for steady-state criteria.

4) The model does not relate to any popular sequential-sampling or drift-diffusion models [119,130]. One interesting future direction of research is to characterize how criterion changes relate to the starting point and speed of the accumulation process posited in these models, which would also allow to predict response time distributions [118].

## Conclusion

In sum, we present a detection-theory model that by design includes the history of stimuli, responses, and outcomes, which together influence upcoming decisions through a single value, the current criterion. Conceptually, the modeling results bring together and demonstrate the importance of two general features of adaptive perceptual decision-making. These are 1) the inclusion of perceptual uncertainty as a factor which modifies the extent of criterion adjustment and 2) the role of differences in relative rather than absolute reward rates to determine steady-state response bias. We further report that these two features are particularly beneficial in non-stationary environments, allowing animals to harvest a larger number of rewards. In the light of these findings, we suggest that future research on the mechanisms of PDM as well as its neuronal underpinnings will benefit from incorporating more frequently and more diverse non-stationary contingencies of reinforcement, as are common in natural environments [131].

## Methods

### Ethics statement

All subjects were kept and treated in accordance with German guidelines for the care and use of animals in science and conducted in agreement with directive 2010/63/EU of the European Parliament. All experimental procedures conducted with rats were approved by the national ethics review board (Landesuntersuchungsamt Rheinland-Pfalz) of the state of Rhineland-Palatinate, Germany (Az. G19-1–094). All experimental procedures conducted with pigeons were approved by the national ethics review board (Landesamt für Natur, Umwelt und Verbraucherschutz) of the state of North Rhine-Westphalia, Germany (Az. 8.87-50.10.37.09.277).

## Subjects

### Rats

Subjects were four male Long Evans rats (Charles River), 6 weeks upon arrival at the institute. Animals were housed in groups of three and lived on a 12-hour reversed day-night cycle (lights off at 8 a.m.). After habituation, they were water-restricted and trained on the behavioral task. Water restriction extended from Sunday to Friday with *ad libitum* water on weekends. On training days, the animals' water intake and body weight were measured daily and supplemental water was provided if necessary, with total water intake adjusted to body weight.

### Pigeons

Subjects were four unsexed domestic pigeons (Columba livia), obtained from local breeders. Pigeons were housed individually in wire-mesh cages inside a temperature- and humidity-controlled colony room with a light period extending from 8 a.m. to 8 p.m. On weekdays, the birds obtained food in the experimental chambers. On weekends, food was freely available in their home cages. The birds' body weight was measured daily, and additional food was provided when an animal's weight dropped below 85% of its free-feeding weight. Water was available ad libitum.

## Apparatus and stimuli

### Rats

Rats were trained in standard operant chambers (ENV-008, Med Associates) placed inside wooden sound-attenuating cubicles whose interior walls were covered with Styrofoam. Each operant chamber featured three conical nose ports. Nose ports were placed along the side wall and equipped with infrared beams to register snout entries. Also, each nose port featured a small well at its bottom which was connected to a pump for delivery of 30 µl of water. A dim house light was on constantly with the exception of time-out punishments (see below).

Sounds were generated in MATLAB (The MathWorks) at a sampling rate of 200 kHz, imported into a custom-written script in Spike2 (Cambridge Electronic Design) which controlled all experimental hardware, then output via a power 1401–3 A/D converter unit (Cambridge Electronic Design) to a conventional stereo amplifier and played by piezoelectric tweeter speakers located at the ceiling of the sound-attenuating cubicle. Following [132], each sound consisted of a chord composed of the sum of 12 logarithmically-spaced pure tones having the same amplitude. Starting from a sound's center frequency (CF), the 12 pure tones spanned the range CF/1.2 to CF*1.2. Sound duration was 100 ms. For each rat, we selected a set of five chords that lay approximately –1.5, –0.5, 0, +0.5 and +1.5 standard deviations from the category boundary on the decision axis (see Table 2 for each rat's stimulus frequencies and further below for an explanation of 'decision axis'). All stimuli were calibrated to 70 dB SPL RMS with a ¼-inch microphone (Microtech Gefell).

Feedback sounds (played for time-out punishments and premature withdrawals) were presented from a different speaker which was attached directly to the wall of the operant box.

### Pigeons

Behavioral testing was carried out in a custom-built operant chamber. The chamber was encased in a sound-attenuating shell; white noise (~80 dB) was presented continuously to mask extraneous sounds. Visual stimuli were presented on a touch screen (Elo 1515L, Tyco Electronics) mounted to one side of the chamber. A computer-controlled custom-built feeder was located centrally beneath the screen. Upon activation, the feeder provided access to a grain reservoir for 0.5-1.5 s (duration was adapted for each animal to ensure adequate food supply depending on body weights and was kept constant throughout the experiment). The chamber was constantly illuminated by two rows of white LEDs positioned beneath the ceiling. Another LED illuminated the food tray during grain delivery. All hardware was controlled by custom-written Matlab code [133].

## Behavioral training and paradigm

### Rats

Rats performed a single-interval forced-choice auditory discrimination task in which they had to categorize a set of chords as belonging to either a high or low frequency category by emitting a response to the left or right choice port, respectively. During task acquisition, correct responses were always rewarded, and incorrect responses were always punished with a time-out of 4 s during which house lights went off. Once animals reached asymptotic performance (approximately 2 months after beginning of initial training), we mapped the chords to the decision axis. The detection theory concept of the decision axis is illustrated in Fig 1 and explained in the main text. Chords were mapped to the decision axis by converting the fraction of R2 responses for each stimulus to a z-score, which gives the distance of the criterion to the stimulus mean (S2 Fig). We then selected, for each subject, five stimuli whose means were located approximately -1.5, 0.5, 0, 0.5 and 1.5 standard deviations from the subjective category boundary (namely, a stimulus that would elicit the same number of R1 as R2). Only these five rat-specific stimuli were used thereafter in the testing phase.

The subjects self-initiated a trial by continuously poking into the center port for a fixed duration (400 ms), after which the stimulus was played. After stimulus offset, they could immediately withdraw from the center port and emit

**Table 2. Stimulus center frequencies (in Hz) of the chords used for each rat.**

| Rat # | Stim 1 | Stim 2 | Stim 3 | Stim 4 | Stim 5 |
|---|---|---|---|---|---|
| 0 | 3249.1 | 5954.5 | 6928.2 | 8061.1 | 14774 |
| 1 | 32491 | 6422.9 | 6928.2 | 7473.2 | 14774 |
| 2 | 3249.1 | 5954.5 | 6928.2 | 8061.1 | 14774 |
| 5 | 2792.4 | 5954.5 | 6928.2 | 8061.1 | 17189 |

an operant response (designated as poking into either the left or the right nose port within 4 s; choice phase). Premature withdrawals (i.e., before stimulus offset) led to trial abort, accompanied by a feedback sound and a time-out of 4 s during which the house light was turned off (with the exception of rat 0 for which the time-out was 0 s). Aborted trials were not repeated and not included for analysis. In most experimental conditions, correct responses were consistently rewarded through delivery of 30 µl of water at the selected response port. All non-rewarded responses, correct or incorrect, were punished with a 4-s time-out during which the house light was switched off and a feedback sound was played. Each session lasted 50 minutes and animals completed between 300 and 600 trials. In sessions where animals did not reach a weight dependent-minimum amount of water, they were supplied the remaining volume by the experimenter.

During the acquisition phase, the task was designed to provide similar numbers of rewards and reward omissions for each of the categories. During the testing phase, however, rats were subjected to several different experimental conditions, in which they experienced category-wise asymmetrical frequencies of rewards and reward omissions. These asymmetries were brought about by manipulating the mapping between stimuli, responses and outcomes (see Fig 2 and further below).

## Pigeons

Subjects were trained on a single-interval forced choice visual categorization task. Fig 7A shows a sketch of the operant chamber, Fig 7B provides details on the behavioral paradigm. Key pecks to three distinct rectangular target areas on the touch screen (henceforth, "pecking keys") were registered as behavioral responses. These three virtual pecking keys were arranged in a horizontal row located about 8 cm above the floor. The central key was positioned in the middle of the monitor, the side keys were placed to the left and the right of the center key. Each trial began with the presentation of an orange-colored rectangle on the center key, accompanied by a 0.5-s pure tone at 1000 Hz. If the animal responded to the stimulus within 3 s (Fig 7B, "Initialize"), one of several discriminative stimuli (shades of gray) replaced the orange rectangle on the center key ("Stimulus"). Failure to peck at the orange key within 3 s aborted the trial; aborted trials were not repeated. Discriminative stimuli were rectangular uniform gray scale images plotted against a uniform black background. The stimuli only differed in terms of their brightness. Gray scale values ranged from 140 to 220, and were selected according to the discrimination capabilities of individual birds (see Table 3 above). The discriminative stimulus was presented for 1 s and then replaced by an orange rectangle. The birds were required to peck at the orange rectangle at least once within 3 s following discriminative stimulus offset to switch off illumination of the center key and trigger the presentation of two orange rectangles on the side keys ("Confirm"). Subjects were required to indicate whether the sample stimulus in the current trial had a gray scale value above or below 180 by pecking at the left or the right choice key, respectively ("Choice"). Correct responses were consistently followed by activation of the feeder for an animal-specific duration in the Baseline, Rich, and Confuse conditions. In the Lean condition, reward occurred only on a fraction of correct trials (see Table 1). Incorrect responses were always followed by a 2-s time-out during which the house light was switched off ("Outcome").

**Table 3.** Grayscale values of the visual stimuli used for each pigeon (monitor grayscale values of 140 and 220 correspond to illuminances of 35 and 76 lux, respectively).

| Pigeon # | Stim 1 | Stim 2 | Stim 3 | Stim 4 | Stim 5 |
|---|---|---|---|---|---|
| **666** | 140 | 165 | 180 | 195 | 220 |
| **850** | 140 | 165 | 180 | 195 | 220 |
| **897** | 150 | 170 | 180 | 190 | 210 |
| **902** | 140 | 165 | 180 | 195 | 220 |

The duration of the inter-trial interval (ITI) was 6 s but was extended whenever the birds pecked at the screen within the last second of the ITI until they refrained from pecking for at least 1 s. Testing sessions were conducted on weekdays. Each session consisted of 280 trials and began with three warm-up trials in which the center key was illuminated orange, and a single key peck triggered food presentation. These trials were not analyzed further.

## Experimental conditions and model predictions

Animals underwent seven different experimental conditions, termed Baseline (B), Rich Left, Rich Right, Lean Left, Lean Right, Confuse Left and Confuse Right. In the experimental condition Rich L, stimulus -1.5 is presented in 50% of trials (C1, consistently associated with R1), while stimulus 0 and stimulus +1.5 (C2) are presented on 25% of trials each and are both associated with R2 (Fig 2). This asymmetry is constructed such that the location of the optimal criterion is to the left of the criterion of the previous symmetric baseline condition. Therefore, an optimal animal should shift its criterion to the left once it enters the new condition (S1 Fig). The same is true for an animal that is mostly driven by reward omissions because most omissions with a criterion of zero follow R1 responses, hence making R1 responses less likely by a criterion shift to the left. Importantly, an Integrate-Rewards account predicts that the criterion should be shifted to the right because most of rewards are also obtained from R1 responses and hence R1 responses will become more likely. The upshot of using this stimulus set is that Integrate-Rewards and Integrate-Reward-Omissions learning models make divergent predictions as to the location of the criterion (shifting to positive and negative values, respectively; see S1.2, predictions for IR and IRO accounts under Rich L conditions). Opposite criterion shifts would be expected if presenting +1.5 in 50% of trials (associated with R2) instead, and presenting -1.5 and 0 in 25% of trials each and reinforcing R1 responses (condition Rich R). See Fig 2 for condition design and S1 for steady state-predictions.

The next condition Confuse L follows a stimulus arrangement in which R1 is reinforced ensuing presentations of -1.5 and +0.5 (C1), while R2 is reinforced if occurring subsequent to presentations of 0 and +1.5 (C2; thus, +0.5 is allocated to the "wrong side" of the category boundary). Again, the Integrate-Rewards- and –Reward-Omissions models make divergent predictions about the direction of the animals' criterion shifts (toward negative values and positive criterion values, respectively). An analogous reasoning applies to condition Confuse R (Figs 2 and S1). Importantly, in both Rich and Confuse conditions, the predictions of each of the two models can be contrasted with an optimization account which predicts criterion shifts in the opposite direction of that of the Integrate-Rewards model in Rich and that of the Integrate-Reward-Omissions model in Confuse. Hence, the two conditions together allow us to diagnose whether animals are more driven by rewards, by reward omissions by both or whether they do something cleverer and can optimize their expected rewards after all.

The last experimental condition considered, Lean, represents a replication of Rich in that the ratio of expected rewards from Category 1 vs Category 2 are the same with the exception that subjects are expected to harvest roughly twice as many rewards in Rich compared to Lean versions of the task. That is because in Lean, unlike the previous conditions, reinforcement is probabilistic, and both non-rewarded correct and incorrect decisions trigger time-out punishment. This design allowed us to test adaptation mechanisms underlying different reward-density contingencies.

The same experimental conditions were similarly run with both rats and pigeons. Each subject (rats and pigeons) stayed in each condition typically for 10 sessions whereas baseline was run typically for 3–5 sessions. Table 1 provides numerical details on all experimental conditions.

## Models

**Integrate Rewards model (IR).** The IR model updates the criterion in a stepwise manner only on rewarded trials. Specifically, the criterion in trial t, c(t), is updated according to the following equation:

$$c(t+1) = \gamma c(t) + \delta(Rew_{R1} - Rew_{R2}).$$

(1)

Here, γ ranges from 0 to 1 (usually, $0.9 < \gamma < 1$) which represents a leaky integration of past criterion values. With $\gamma = 1$ (i.e., no leakage), the criterion quickly drifts to infinity (see [67,134]). δ is a learning rate parameter controlling the size of the criterion adjustment. Following a reward for R1 ($Rew_{R1} = 1$, $Rew_{R2} = 0$), or for R2 ($Rew_{R1} = 0$, $Rew_{R2} = 1$), the criterion shifts such that the subject is more likely to choose the same response again in the next trial that led to reinforcement in the current trial.

In the steady state, the criterion position of this model depends on the absolute difference in reinforcement obtained from R1 and R2. To understand why, let us derive this criterion position. The model is in an equilibrium when the criterion will not change on average, i.e., $c(t) = \mathbb{E}[c(t+1)]$. We can compute $\mathbb{E}[c(t+1)]$ by averaging over all possible outcomes of a trial, weighted with their probability to occur:

$$\mathbb{E}[c(t+1)] = \gamma c(t) * P[NoRew] + (\gamma c(t) + \delta) * P[Rew_{R1}] + (\gamma c(t) - \delta) * P[Rew_{R2}] = \gamma c(t) + \delta * (P[Rew_{R1}] + P[Rew_{R2}]) \cdot \quad (2)$$

The probability of receiving a reward for a certain response, e.g., $P[Rew_{R1}]$, is the same as the expected reinforcement in a trial, $\mathbb{E}[Rew_{R1}]$, therefore we can use both terms interchangeably. So, in the equilibrium

$$c(t) = \gamma c(t) + \delta \left( \mathbb{E}[Rew_{R1}] - \mathbb{E}[Rew_{R2}] \right), \quad (3)$$

which can be rearranged to

$$c(t) = \frac{\delta}{1 - \gamma} \left( \mathbb{E}[Rew_{R1}] - \mathbb{E}[Rew_{R2}] \right). \quad (4)$$

The expected values in this equation are straightforward to determine for a given c. To determine the steady-state criterion, we thus calculated the solution to this steady-state equation through numerical optimization. A graphical example is shown in S1 Fig.

**Integrate Rewards model with convergence to a steady state defined by Relative Differences (IR-RD).** This model version follows the same learning design as IR but features no update (including no leak) in the absence of reward.

If $Rew_{R1} = 1$ or $Rew_{R2} = 1$

$$c(t+1) = \gamma c(t) + \delta(Rew_{R1} - Rew_{R2}), \quad (5)$$

else

$$c(t+1) = c(t). \quad (6)$$

By including this modification, it can be demonstrated that the criterion in the steady state will depend on the difference between relative instead of absolute reward rates. This particular feature makes it more consistent with the matching law, one of the most widely observed equilibriums in decision-making.

As before, we can compute the criterion position for the steady state by averaging over all possible outcomes of a trial to get $\mathbb{E}[c(t+1)]$ and then set $c(t) = \mathbb{E}[c(t+1)]$ to determine the equilibrium.

$$\begin{aligned}
\mathbb{E}[c(t+1)] &= c(t) * P[NoRew] + (\gamma c(t) + \delta) * P[Rew_{R1}] + (\gamma c(t) - \delta) * P[Rew_{R2}] \\
&= c(t) * (1 - P[Rew_{R1}] - P[Rew_{R2}] + \gamma P[Rew_{R1}] + \gamma P[Rew_{R2}]) \\
&\quad + \delta \left( P[Rew_{R1}] - P[Rew_{R2}] \right) \\
&= c(t) * (1 - (1 - \gamma) \left( P[Rew_{R1}] + P[Rew_{R2}] \right)) + \delta(P[Rew_{R1}] - P[Rew_{R2}])
\end{aligned}$$

$$(7)$$

Replacing P[Rew] with with $\mathbb{E}$[Rew] and using the equilibrium condition thus gives

$$c(t) = c(t) * \left(1 - (1-\gamma)\left(\mathbb{E}\left[Rew_{R1}\right] + \mathbb{E}\left[Rew_{R2}\right]\right)\right) + \delta\left(\mathbb{E}\left[Rew_{R1}\right] - \mathbb{E}\left[Rew_{R2}\right]\right),\tag{8}$$

which can be rearranged to

$$c(t) = \frac{\delta}{1-\gamma}\left(\frac{\mathbb{E}\left[Rew_{R1}\right] - \mathbb{E}\left[Rew_{R2}\right]}{\mathbb{E}\left[Rew_{R1}\right] + \mathbb{E}\left[Rew_{R2}\right]}\right).\tag{9}$$

Here, the absolute difference in reinforcement is scaled by the total amount of reinforcement, thereby, removing the dependence on the reward density.

**Integrate Rewards model with Stimulus-specific Learning Rates (IR-SLR).** This model is identical to the IR model, with the difference that the learning rate parameter $\delta$ varies for each individual stimulus. The rationale for using stimulus-specific learning parameters is the hypothesis that animals learn less from events which are more certainly predicted on the basis of sensory evidence. The algorithm is

$$c(t+1) = \gamma c(t) + \delta_{stim}(Rew_{R1} - Rew_{R2}),\tag{10}$$

where $\delta_{stim}$ is the stimulus-specific value of $\delta$ for the stimulus that was presented in trial t. Since there were five stimuli over all experiments, there are five values of $\delta_{stim}$.

**Integrate Rewards model with Stimulus-specific Learning Rates and with convergence to a steady state defined by Relative Differences (IR-SLR-RD).** As with the IR model the criterion in the IR-SLR model can be designed to converge to a steady state governed by the relative instead of absolute reward differences, in line with the matching law.

If $Rew_{R1}$=1 or $Rew_{R2}$=1:

$$c(t+1) = \gamma c(t) + \delta_{stim}\left(Rew_{R1} - Rew_{R2}\right).\tag{11}$$

otherwise, the criterion remained unaffected (see equation 6).

**Integrate Rewards model with a reduced number Stimulus-specific Learning Rates (IR-SLR(red)).** This model version is implemented because in pigeons the full IR-SLR model does not provide satisfactory simulations (in 2 out of the 4 animals the simulations go to extreme choice behavior in Confuse L and Confuse R conditions (see Fig S6B and S6D)). This is because unlike in the rat experiment, in the pigeon experiment the learning rates for stimulus 2 and stimulus 4 are fitted only on the base of behavior in Confuse conditions. As shown by the simulations, for some subjects our models are unable to reproduce animals' behavior in this condition (also see treatment in Discussion). As a result, the learning rates of stimulus 2 and 4 are unreliable. To tackle this, we built a reduced model that we applied to all pigeon datasets, which has only two learning rates, one associated with easy stimuli (1 and 5) and a second learning rate associated with difficult stimuli (2, 3 and 4). This reduced stimulus-specific learning rate modulation is referred to in the manuscript as SLR(red) and does not qualitatively alter model performance although it features less degrees of freedom (see Figs 7G and S7E).

**Integrate Reward Omissions model (IRO).** The IRO model follows the same logic as the IR model, the main difference being that it updates the criterion only on unrewarded trials. For reward omissions following R1, $NoRew_{R1} = 1$ and $NoRew_{R2} = 0$, and vice versa reward omissions following R2, i.e., $NoRew_{R1} = 0$ and $NoRew_{R2} = 1$. The model renders the unrewarded response less likely to occur in the following trial by shifting the criterion as described by:

$$c(t+1) = \gamma c(t) + \upsilon \left( NoRew_{R2} - NoRew_{R1} \right). \tag{12}$$

The size of the criterion step is now controlled by the learning parameter $\upsilon$. Negative learning rates imply a tendency of the model to increase the probability to choose the response that leads to a reward omission.

**Integrate Rewards & Reward Omissions model (IR&RO).** The IR&RO model updates the criterion in both trials with rewards and reward omissions according to:

$$c(t+1) = \gamma c(t) + \delta \left( Rew_{R1} - Rew_{R2} \right) + \upsilon(NoRew_{R2} - NoRew_{R1}). \tag{13}$$

The size of the criterion step after rewards is controlled by $\delta$ whereas after reward omission it is controlled by $\upsilon$.

The IRO and IR&RO models were expanded to encompass stimulus-specific learning and/or converge to a steady state defined by relative differences, but as shown in S5A Fig (rats) and S7E Fig (pigeons), the fits still yielded negative learning rates and were therefore not considered in the main body of the paper.

**Criterion setting according to an optimal account.** To benchmark the animals' performance, we computed the optimal location of the criterion within the SDT framework as a function of the fitted stimulus distribution means and the experimental reward contingencies for each condition. The optimal criterion maximizes the expected reward

$$\begin{aligned} P\left(Rew|c\right) &= P\left(Rew|c, C1\right) \times P(C1) + P\left(Rew|c, C2\right) \times P(C2) \\ &= P\left(Rew|C1, R1\right) \times P\left(R1|c, C1\right) \times P(C1) + P\left(Rew|C2, R2\right) \times P\left(R2|c, C2\right) \times P(C2) \\ &= P\left(Rew|C1, R1\right) \times P\left(X < c|C1\right) \times P(C1) + P(Rew|C2, R2) \times \left(1 - P\left(X < c|C2\right)\right) \times P(C2). \end{aligned} \tag{14}$$

To maximize this, the first derivative with respect to c needs to be zero, which is equivalent to

$$P\left(Rew|C1, R1\right) \times p\left(X = c|C1\right) \times P(C1) = P(Rew|C2, R2) \times p(X = c|C2) \times P(C2). \tag{15}$$

We solved this equation through numerical optimization. Conceptually, it means that the optimal criterion is located at the intersection of the category distributions scaled with the presentation and reward probability (see S1 Fig and Fig 2A for visualization). Note that the scaled category distribution can be obtained by summing the scaled stimulus distributions for all stimuli that belong to that category, i.e., for category 1,

$$p\left(X = c|C1\right) \times P(C1) = \sum_{i \in \{C1\}}^{n} p(X = c|S_i) \times P(S_i). \tag{16}$$

We use the term 'decision distribution' as a shorthand for the scaled category distribution because it summarizes all three decision factors (x, P(Rew), and P(S)) and can be interpreted as the function of observation-dependent action values, scaled by the probability of the respective observation.

**One-criterion-per-session model (OCPS).** Because we manipulated the stimulus presentation probabilities (fraction of trials that belonged to each category) across conditions, the R2 probabilities do not directly reflect the decision criterion but partly follow a different trajectory. We therefore show the behavioral data not only as P(R2) but also as criterion. In contrast to P(R2), the criterion is not directly observable from the behavior. To determine the criterion, we modeled the animals' stimulus-wise R2 probabilities for each session as a function of stimulus means (one for each stimulus, remaining constant over the course of the experiment) and a session-specific criterion.

Using the standard signal detection theory model and assuming a fixed criterion for each session, the probability of choosing R2 in session $j$ when stimulus $i$ was presented can be expressed as

$$P_j(R2|S_i) = \Phi(\mu_i - c_j),\tag{17}$$

where $\Phi$ denotes the cumulative normal distribution, $\mu_i$ is the mean of the distribution of stimulus i and $c_j$ is the criterion in session j. In our one-criterion-per-session model, we thus computed the z-scored probabilities of responding R2 for each stimulus i and session j, $d_{ij}$, by taking the inverse cumulative normal distribution $\Phi^{-1}$ of the observed response probabilities:

$$d_{ij} = \Phi^{-1}(P_j(R2|S_i)) = \mu_i - c_j.\tag{18}$$

This gives us a model that has one parameter $\mu_i$ per stimulus and one parameter $c_j$ per session, representing the criterion for each respective session, which we fitted with linear least-squares regression, using dummy coding for the stimuli and sessions.

**Logistic regression analysis.** We built a logistic regression model (using the glmfit function in Matlab, assuming binomially distributed responses and using the logit link function) to investigate the impact of past rewards and reward omissions on the subsequent choice probabilities in rats. We regressed the influence of previous rewards (Rew) and reward omissions (NoRew) in trials t-1, t-2, t-3, and t-4 on the response (0 for R1/ 1 for R2) in trial t. All the history regressors were built so that the propensity towards R2 was coded as 1 and towards R1 as -1. Specifically, both reward for R1 and no reward for R2 were coded as a -1 in the Rew and NoRew regressors respectively, whereas reward for R2 and no reward for R1 were both coded as +1. We additionally included dummy-coded regressors for each stimulus i (one regressor per stimulus), as well as one regressor for each session k to account for slow criterion changes (excluding the first session to avoid collinearity of regressors). Only after completion of the regressor table, we excluded aborted trials from analysis, thereby removing them from the dependent variable, but still having an indirect impact in the history regressors (e.g., if an aborted trial happened at t-1, and although that given trial would not be considered, $Rew_{t-1}$ and $NoRew_{t-1}$ regressors would incidentally both equal 0 at row t).

$$log\left(\frac{P_{R2}}{P_{R1}}\right) = \beta_0 + \sum_{i=1}^{5}\beta_i^{Stim}Stim_i + \sum_{j=1}^{4}\beta_j^{Rew}Rew_{t-j} + \sum_{j=1}^{4}\beta_j^{NoRew}NoRew_{t-j} + \sum_{k=2}^{total}\beta_k^{Sess}Sess_k\tag{19}$$

## Model fitting and forward simulations

The model fitting was performed as described in [68]. To summarize, for a fixed leak parameter, the models can be expressed as a generalized linear model. [135] shows that the likelihood function of these models has a unique maximum which can therefore be found using standard numerical optimization methods. The models were fitted by repeating this procedure for different values of $\gamma$ and choosing the parameters leading to the overall maximum likelihood.

We compared the goodness of fit of the different criterion learning models through calculations and comparison of the Bayesian Information Criterion (BIC) values for each of the respective model fits:

$$BIC = [2 * NLL + k * log(N)]\tag{20}$$

where *k* is the number of free parameters of the model, N the number of trials and NLL is the negative log likelihood of the observed data given the model and fitted parameters. The advantage of using the BIC for model comparison rather than NLL is that the BIC controls for the number of free parameters, which may differ between models. Models with smaller BIC values are preferred. According to [136], the strength of evidence against the model with the higher BIC value is "positive" for BIC differences of 2–6, "strong" for differences of 6–10, and "very strong" for larger differences.

In each simulation, the models were presented with a newly generated sequence of stimuli and potential rewards (i.e., sequence of trials where a reward will be acquired given a correct response), which was obtained from shuffling all the trials within the same condition. For each trial, the model's response was sampled according to its predicted probability for each response, and then the criterion was updated according to the outcome following the respective model's update rule. For each animal and model, we ran 1000 simulations and computed the mean and standard deviation of the fraction of R2 responses per session.

Note that in order to run simulations with models featuring SLR modulations in pigeons we had to implement a *reduced* form with only 2 rather than 5 fitted learning rates (coined *IR-SLR(red)* model, see Models section above). The results with 2 or 5 learning rates provided comparably similar improvements as depicted by the BICs differences (Figs 7F and S7C and justification of this approach above in the Models section).

## Supporting information

**S1 Fig. Exemplary predictions of steady-state criterion locations for the three initially considered models for all experimental conditions.** Each row shows the predicted criterion locations (vertical colored lines) of the optimal account, IR, IRO and IR&RO accounts for a certain experimental condition. In the first column, the optimal account is shown. In all plots in this column, the green curve is the objective reward function, which represents the total expected probability of reinforcement in a trial dependent on the criterion position. The black and gray lines are the decision distributions for R1- and R2-associated stimulus categories, respectively. Optimal performance is achieved at the maximum of the objective reward function; the corresponding criterion is plotted as a green vertical line. In the other columns, predictions for the IR, IRO and IR&RO models are shown. The colored lines depict the category-wise differences between the expected probabilities for reward (IR), reward omissions (IRO), or both (IR&RO). The black and gray lines depict the same, but conditioned on a trial with R1 and a trial with R2, respectively. Additionally, a dashed black line through zero is plotted, whose slope depends on the leakage term $\gamma$ and the step size $\delta$ or $\upsilon$: $(1-\gamma)/\delta$ for the IR model, $(1-\gamma)/\upsilon$ for the IRO model, and $(1-\gamma)/\delta=(1-\gamma)/\upsilon$ for the IR&RO model. The predicted criterion location for the models is at the intersection of this straight line with the colored line (see Methods section for the derivation). Parameter values: $\gamma=0.99$, $\delta=\upsilon=0.04$.
(EPS)

**S2 Fig. Subject-specific construction of stimulus sets and validation of SDT framework. a.** Psychometric functions of each rat, fitted with logistic functions and allowing for lapses (orange). Each blue data point represents the fraction of leftward (R2) choices for one stimulus. Bars represent standard errors of the means. **b.** Same as a, but after z-scoring P(R2) values to map the stimuli to perceptual space (here, 0 corresponds to the stimulus for which the subject will respond R1 or R2 with equal probability). This mapping was carried out to select a suitable set of stimuli that would match as closely as possible the location of the stimulus means in the perceptual space specified by the experimental design (-1.5, -0.5, 0, 0.5, 1.5). **c.** Scatterplots of measured criterion location against criterion locations reconstructed with the OCPS model. Each data point represents a single subject's choice probability for one stimulus in a single session.
(EPS)

**S3 Fig. Steady-state reward densities and stimulus means. a.** Data points represent the steady-state reward densities (i.e., average rewards per trial) computed over the last 3 sessions of each condition for each animal, whereas crosses represent means across all subjects. **b.** Individual stimulus means fitted by the OCPS model as a function of stimulus number. Individual points represent measured values for each subject across the entire experiment while crosses represent means across subjects. Dashed lines reference the intended values -1.5, -0.5, 0, 0.5, and 1.5.
(EPS)

**S4 Fig. Individual steady-state vs. optimal criteria in the experimental conditions.** Using the stimulus means fitted for each rat through the OCPS model, the two decision distributions (black lines) as well as the objective reward functions

(ORF) can be calculated. The ORF is plotted in green and its maximum, shown as a vertical green line, indicates the reward-maximizing criterion. The vertical black lines mark the animals' steady-state criterion in the respective condition. (EPS)

**S5 Fig. Parameter fits and model comparison for models involving learning from reward omissions. a.** υ values for all model versions that feature learning after reward omissions. Both the standard IRO (purple) and IR&RO (yellow) models were extended to feature five, rather than one, stimulus-specific learning rates (SLR). Small crosses represent individual subjects' fitted υ parameters, whereas thick crosses represent means over the four subjects. With very few exceptions, υ values turned out negative. **b.** Boxplots of BIC values for the fits of the three initially considered models IR, IRO and IR&RO.
(EPS)

**S6 Fig. Response bias (P(R2)), model fits to P(R2) and criterion, and simulations for all subjects and conditions.** Individual fits (visualized as P(R2) and criterion) and simulations of different model versions considered in the main body of the manuscript for all rats and pigeons. **a.** IR model. **b.** IR-SLR model. **c.** IR-RD model. **d.** IR-SLR-RD model. **e.** IRO model. **f.** IR&RO model. **g.** IR-SLR(red). This model version features only two (instead of five) learning rates and is applied to pigeons only. **h.** IR-SLR(red)-RD, pigeons only, as in g.
(PDF)

**S7 Fig. Additional results from the pigeon experiment. a.** Session-wise fraction of R2 responses (black) and number of rewards per trial (green) for each pigeon. Gray shading indicates baseline sessions. **b.** Development of hit rate (HR, blue) for stimulus 1 and false alarm rate (FA, red) for stimulus 5 over the course of behavioral testing. Thin lines represent data from individual subjects, thick lines represent the means over the 4 subjects. **c.** Individual stimulus means fitted by the OCPS model as a function of stimulus number. Data points represent individual values while crosses represent means across subjects. Dotted lines reference the intended theoretical values. **d.** Absolute BIC values for the three initially considered models. **e.** BIC values of the competitor models after subtracting the BIC of the full model. Unlike the full model (IR-SLR(red)-RD) used in Fig 7, which features only 2 learning rates, these BICs result from a full model version which, as in rats, features 5 learning rates. In pigeons, the model with 5 learning rates systematically leads to extreme-choice behavior in Confuse conditions (see S6B and S6D Fig, Pigeons 850 & 902). The full model with only two learning rates is able to fit and reproduce the data similarly and its usage leads to qualitatively identical conclusions. The three median ΔBIC values are 1691 (IR), 1697 (IR-RD), and 84.79 (IR-SLR). **f.** As in S5 Fig but for pigeons, υ values for all model versions that feature learning after reward omissions. Purple and yellow refer to IRO and IR&RO model versions not included in the main text.
(EPS)

## Acknowledgments

The authors thank Alex Hyafil for the assistance with the logistic regression model used to investigate the influence of rewards and reward omissions on subsequent choices.

## Author contributions

**Conceptualization:** Luis de la Cuesta-Ferrer, Frank Jäkel, Maik Christopher Stüttgen.

**Data curation:** Luis de la Cuesta-Ferrer, Maik Christopher Stüttgen.

**Formal analysis:** Luis de la Cuesta-Ferrer, Christina Koß, Frank Jäkel.

**Funding acquisition:** Frank Jäkel, Maik Christopher Stüttgen.

**Investigation:** Luis de la Cuesta-Ferrer, Sarah Starosta, Nils Kasties, Daniel Lengersdorf, Frank Jäkel, Maik Christopher Stüttgen.

**Methodology:** Luis de la Cuesta-Ferrer, Christina Koß, Sarah Starosta, Nils Kasties, Daniel Lengersdorf, Frank Jäkel, Maik Christopher Stüttgen.

**Project administration:** Frank Jäkel, Maik Christopher Stüttgen.

**Software:** Luis de la Cuesta-Ferrer, Christina Koß, Frank Jäkel.

**Supervision:** Frank Jäkel, Maik Christopher Stüttgen.

**Validation:** Luis de la Cuesta-Ferrer, Frank Jäkel.

**Visualization:** Luis de la Cuesta-Ferrer, Christina Koß.

**Writing – original draft:** Luis de la Cuesta-Ferrer, Christina Koß, Maik Christopher Stüttgen.

**Writing – review & editing:** Luis de la Cuesta-Ferrer, Christina Koß, Frank Jäkel, Maik Christopher Stüttgen.

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
