## [Decision Letter · Decision Letter 0]

14 Jan 2025

PCOMPBIOL-D-24-01968

Stimulus uncertainty and relative reward rates determine adaptive responding in perceptual decision-making

PLOS Computational Biology

Dear Dr. Stüttgen,

Thank you for submitting your manuscript to PLOS Computational Biology. After careful consideration, we feel that it has merit but does not fully meet PLOS Computational Biology's publication criteria as it currently stands. Therefore, we invite you to submit a revised version of the manuscript that addresses the points raised during the review process.

Specifically, both reviewers raised concerns about the clarity of the experimental design, the goals of the study, and its connections to existing literature (e.g., studies on matching experiments). Additionally, several statements in the Results section need to be supported with appropriate statistical tests. To strengthen the connection to previous literature, it would be helpful to include a discussion and references to studies examining the effects of global reward rate (e.g., Wittmann et al., Nature Communications, 2020) and reward uncertainty (Woo et al., Cognitive, Affective, & Behavioral Neuroscience, 2023) on learning and decision making. Lastly, please ensure precise language when discussing insight your study provides into the mechanisms underlying perceptual decision-making.

Please submit your revised manuscript within 60 days Mar 16 2025 11:59PM. If you will need more time than this to complete your revisions, please reply to this message or contact the journal office at ploscompbiol@plos.org. Please include the following items when submitting your revised manuscript:

We look forward to receiving your revised manuscript.

Kind regards,

Alireza Soltani

Academic Editor

PLOS Computational Biology

Marieke van Vugt

Section Editor

PLOS Computational Biology

**Journal Requirements:**

3) We notice that your supplementary Figures, and Tables are included in the manuscript file. Please remove them and upload them with the file type 'Supporting Information'. Please ensure that each Supporting Information file has a legend listed in the manuscript after the references list.

Potential Copyright Issues:

- Figures 2A and 6A; Please confirm whether you drew the images / clip-art within the figure panels by hand. If you did not draw the images, please provide a link to the source of the images or icons and their license / terms of use; or written permission from the copyright holder to publish the images or icons under our CC BY 4.0 license. Alternatively, you may replace the images with open source alternatives. See these open source resources you may use to replace images / clip-art:

**Reviewers' comments:**

Reviewer's Responses to Questions

**Comments to the Authors:**

Reviewer #1: Cuesta-Ferrer et al. investigate decision-making and its main determinants in pigeons and rats using a perceptual decision-making task (PDM). The authors manipulate reward probabilities, stimulus presentation probabilities, and discrimination difficulty, employing detection theory-based models to explain the behavior. They demonstrate that obtained rewards, rather than reward omissions, drive adaptation in behavior in response to changes in contingencies. While the manuscript addresses an important question, reliance on supplementary information for crucial experiment details and the lack of a comprehensive statistical report at many points hindered my ability to fully appreciate the significance of the findings and the motivation behind the study.

Major comments:

1) The topic of the manuscript has been widely studied in the field and will continue to be explored in the future. However, the introduction section of the paper does not adequately reflect this. Specifically, there should be a more thorough literature review on the models that utilize signal detection theory and confidence to explain adaptive behavior. Additionally, the paper should clarify the gaps in the existing literature that the authors aim to address, as the motivation for the study is currently unclear.

2) The abstract starts with "In an ever-changing environment, animals must learn to be flexible." This suggests that learning and reversals in learning are necessary. However, based on the methods section, all tests occurred after the animals became fully familiar with the task, and there was no reversal in the S-R map implemented.

3) What is the neuroscience intuition behind changing the criteria thresholds in Fig 1, particularly in instances where S1 had an R2 response, and S2 had an R1 response? Additionally, the choice of a rewarding option relies on both perceptual acuity and an understanding of the rules. Consequently, moving the criterion line will attribute the reward solely to perception, which can lead to issues with reward assignment.

4) The equations in Fig. 1b do not have a parameter indicating what the stimulus was. This is probably a notation issue, though.

5) How do the authors think the model works if there was a reversal in rule (S-R mapping)?

6) There are several important details, such as reward probabilities, that are referred to as supplementary results, but they should be clearly written in the main text rather than supplementary information.

7) The experiment description for pigeons could use some revision to enhance comprehension. Figure 6.a and 6.b are too abstract, making it difficult to understand the experiment based on the description in the methods section.

8) There are many instances of statements in the results section that require a report of statistics, but nothing is reported (e.g., 4.3, 4.4, and 4.5).

Reviewer #2: This paper investigated decision making in rats and pigeons under different task contingencies that were determined based on reward probabilities, stimulus probabilities, and stimulus discriminability. Crucially, authors took account of trial by trial adjustments of decision thresholds under the signal detection theory framework. Authors tested three different models: reward, reward omission, and both. They find that the integration of the rewards considering stimulus difficulty and reward difference metrics (i.e., relative difference) best accounts for their data. Finally, the authors find that the performance of rats and pigeons were comparable in terms of model fits and that these animals nearly optimized their decisions (reward maximization). I find the paper very interesting and well-written. I have minor comments.

1- The paper seems to have overlooked a large set of literature that directly relates to the current work. I think the inclusion of these studies (primarily by the Balci group) to the paper is necessary given their direct relevance not only in terms of the results but also the research questions and the theoretical approach (e.g., statistical decision theory considering stimulus uncertainty, optimality).

- Balci, F., Freestone, D., & Gallistel, C. R. (2009). Risk assessment in man and mouse. Proceedings of the National Academy of Sciences of the United States of America, 106(7), 2459–2463. https://doi.org/10.1073/pnas.0812709106

- Tosun, T., Gür, E., & Balcı, F. (2016). Mice plan decision strategies based on previously learned time intervals, locations, and probabilities. Proceedings of the National Academy of Sciences of the United States of America, 113(3), 787–792. https://doi.org/10.1073/pnas.1518316113

- Akdoğan, B., & Balcı, F. (2016). Stimulus probability effects on temporal bisection performance of mice (Mus musculus). Animal cognition, 19(1), 15–30. https://doi.org/10.1007/s10071-015-0909-6

- Gür, E., Duyan, Y. A., & Balcı, F. (2019). Probabilistic Information Modulates the Timed Response Inhibition Deficit in Aging Mice. Frontiers in behavioral neuroscience, 13, 196. https://doi.org/10.3389/fnbeh.2019.00196

This is not a comprehensive list. I suggest the authors to look into these work.

2- The visual inspections of the adjustments point at a very abrupt and near immediate adjustments, which also favors representational and computational accounts (see also Tosun et al., 2016 listed above). To this end, I suggest authors to also look at the following papers:

- Kheifets, A., & Gallistel, C. R. (2012). Mice take calculated risks. Proceedings of the National Academy of Sciences of the United States of America, 109(22), 8776–8779. https://doi.org/10.1073/pnas.1205131109

- Gallistel, C. R., King, A. P., Gottlieb, D., Balci, F., Papachristos, E. B., Szalecki, M., & Carbone, K. S. (2007). Is matching innate?. Journal of the experimental analysis of behavior, 87(2), 161–199. https://doi.org/10.1901/jeab.2007.92-05

The last paper I listed is particularly relevant in consideration of the matching law that the authors mention in the paper.

- Finally, albeit the authors offer a "mechanistic" approach, their approach is still descriptive. A fully generative approach should account for the behaviors in their full complexity (e.g., drift diffusion model), which includes response times in the authors' work.

Overall, this is a beautiful paper, which addresses an important research question.

**Have the authors made all data and (if applicable) computational code underlying the findings in their manuscript fully available?**

Reviewer #1: Yes

Reviewer #2: Yes

PLOS authors have the option to publish the peer review history of their article (what does this mean? ). If published, this will include your full peer review and any attached files.

**Do you want your identity to be public for this peer review?** For information about this choice, including consent withdrawal, please see our Privacy Policy .

Reviewer #1: No

Reviewer #2: No

**Figure resubmission:**

**Reproducibility:**



---

## [Decision Letter · Decision Letter 1]

19 Apr 2025

Dear Dr. Stüttgen,

We are pleased to inform you that your manuscript 'Stimulus uncertainty and relative reward rates determine adaptive responding in perceptual decision-making' has been provisionally accepted for publication in PLOS Computational Biology.

Best regards,

Alireza Soltani

Academic Editor

PLOS Computational Biology

Marieke van Vugt

Section Editor

PLOS Computational Biology

Reviewer's Responses to Questions

**Comments to the Authors:**

Reviewer #1: The authors have fully addressed the concerns and comments.

Reviewer #2: Authors have done a sufficiently good job in revising the manuscript and responding to my questions.

**Have the authors made all data and (if applicable) computational code underlying the findings in their manuscript fully available?**

Reviewer #1: Yes

Reviewer #2: Yes

PLOS authors have the option to publish the peer review history of their article (what does this mean? ). If published, this will include your full peer review and any attached files.

**Do you want your identity to be public for this peer review?** For information about this choice, including consent withdrawal, please see our Privacy Policy .

Reviewer #1: No

Reviewer #2: No

---

## [Editor Report · Acceptance letter]

PCOMPBIOL-D-24-01968R1

Stimulus uncertainty and relative reward rates determine adaptive responding in perceptual decision-making

Dear Dr Stüttgen,

I am pleased to inform you that your manuscript has been formally accepted for publication in PLOS Computational Biology. Your manuscript is now with our production department and you will be notified of the publication date in due course.

With kind regards,

Anita Estes
